# Plasmonic nanoparticle amyloid corona for screening Aβ oligomeric aggregate-degrading drugs

Dongtak Lee [1], Dongsung Park[1,2], Insu Kim [1], Sang Won Lee[1], Wonseok Lee[3], Kyo Seon Hwang[2], Jeong Hoon Lee [4,8 ✉], Gyudo Lee[5,6,8 ✉] & Dae Sung Yoon [1,7,8 ✉]

The generation of toxic amyloid β (Aβ) oligomers is a central feature of the onset and progression of Alzheimer's disease (AD). Drug discoveries for Aβ oligomer degradation have been hampered by the difficulty of Aβ oligomer purification and a lack of screening tools. Here, we report a plasmonic nanoparticle amyloid corona (PNAC) for quantifying the efficacy of Aβ oligomeric aggregate-degrading drugs. Our strategy is to monitor the drug-induced degradation of oligomeric aggregates by analyzing the colorimetric responses of PNACs. To test our strategy, we use Aβ-degrading proteases (protease XIV and MMP-9) and subsequently various small-molecule substances that have shown benefits in the treatment of AD. We demonstrate that this strategy with PNAC can identify effective drugs for eliminating oligomeric aggregates. Thus, this approach presents an appealing opportunity to reduce attrition problems in drug discovery for AD treatment.

[1] School of Biomedical Engineering, Korea University, Seoul 02841, South Korea. [2] Department of Clinical Pharmacology and Therapeutics, College of Medicine, Kyung Hee University, Seoul 02447, South Korea. [3] Department of Control and Instrumentation Engineering, Korea University, Sejong 30019, South Korea. [4] Department of Electrical Engineering, Kwangwoon University, Seoul 01897, South Korea. [5] Department of Biotechnology and Bioinformatics, Korea University, Sejong 30019, South Korea. [6] Interdisciplinary Graduate Program for Artificial Intelligence Smart Convergence Technology, Korea University, Sejong 30019, South Korea. [7] Interdisciplinary Program in Precision Public Health, Korea University, Seoul 02841, South Korea. [8] These authors jointly supervised this work: Jeong Hoon Lee, Gyudo Lee, Dae Sung Yoon. ✉email: jhlee@kw.ac.kr; lkd0807@korea.ac.kr; dsyoon@korea.ac.kr

Alzheimer's disease (AD) is an incurable neurodegenerative disease that imposes significant social and economic costs worldwide[1]. According to the amyloid hypothesis, the aggregation of amyloid β (Aβ) initiates a cascade of molecular events that eventually lead to AD[2]. Recently, increasing evidence has implicated Aβ oligomers, rather than mature Aβ fibrils and plaques[3–5], as the major agents responsible for cellular toxicity[6,7], synaptic dysfunction[8], and tau hyperphosphorylation[9,10]. To date, many studies researching the prevention of the formation of Aβ aggregates, including Aβ oligomers, have used molecular chaperones[11], small molecules[12,13], and antibodies[14]. Although these approaches can effectively prevent the additional formation of Aβ oligomers, existing Aβ oligomers remain active in the brain and cause secondary nucleation in AD progression[15,16]. Therefore, the most desirable anti-Aβ drugs would eliminate existing Aβ oligomers in the brains of AD patients.

Many researchers have sought to discover anti-Aβ drugs using conventional in vitro screening methods, such as fluorescent spectroscopy with chemical dyes (e.g., thioflavin T, Congo red, 8-anilino-1-naphthalenesulfonic acid [ANS], and bis-ANS)[13,17], cytotoxicity assays with induced pluripotent stem cell-derived (iPSC-derived) neurons[18], and mass spectroscopy[19]. However, these methods are limited in capacity for the rapid screening of various reagents by their long required incubation times, complex pretreatment steps, high sample volume requirements, and high cost[16]. In particular, fluorescent spectroscopy-based screening methods are restricted to the investigation of the pharmacokinetic properties of Aβ oligomer-targeting drugs because of the lack of exclusive fluorescent probes for Aβ oligomers[17]. Furthermore, no purification technique exists that yields only the oligomer form without monomers or fibrils; this hinders the accurate assessment of the efficacy of Aβ oligomer-targeting drugs[20].

To address these problems, herein, we fabricated a plasmonic nanoparticle amyloid corona (PNAC) where the corona of a single gold nanoparticle (AuNP) comprises oligomeric aggregates (Fig. 1a). Using these PNACs, drug screening was performed; the main strategy for monitoring amyloid corona degradation is illustrated in Fig. 1e. As an effective drug degrades the oligomeric aggregates in the PNACs over time, the amyloid corona is removed from the PNAC surfaces; consequently, the PNACs that have lost their amyloid corona become aggregated (Supplementary Figs. 1 and 2), causing a color change of the PNAC solution from red to purple. With ineffective drugs, the amyloid corona on the PNACs remains intact, causing no color changes. We believe that the proposed platform combining PNAC synthesis and colorimetric drug screening can contribute to the discovery and pharmacokinetic analysis of Aβ-related drugs.

## Results and discussion

**Ligand-free amyloid corona formation on AuNPs.** First, we fabricated PNACs using variously sized AuNPs (i.e., 20-, 50-, and 100 nm), which were synthesized via citrate reduction (Supplementary Fig. 3). The negatively charged AuNPs electrostatically interacted with the N terminal of the Aβ monomer[21], which markedly accelerated the formation of amyloids on the AuNP surface. This acceleration occurred because the AuNPs increased the local peptide concentration by electrostatic and solvation forces, thus facilitating the formation of a secondary ordered Aβ structure[22]. Among the PNACs with various sizes (20-, 50-, and 100-nm PNACs), we found that the 20-nm PNACs showed the greatest uniformity and steric stability. In gel electrophoresis (Supplementary Fig. 4), the 20-nm PNACs exhibited the narrowest band, implying that they shared the same physicochemical features (excellent uniformity and stability). Unlike the 20-nm PNACs, the 50- and 100-nm PNACs were unstable in phosphate-buffered

saline (PBS), as shown in Supplementary Fig. 5. Using high-resolution transmission electron microscopy (HRTEM), we observed the formation process of the amyloid corona over time (Fig. 1b). After the Aβ oligomerization process, the amyloid corona is observed on the AuNPs. Initially, the Aβ monomers react with each other (i.e., primary nucleation) and begin to cover the AuNP surface. With time, the Aβ shell is thickened (Fig. 1b, panel 3), eventually forming an amyloid corona of oligomeric aggregates on the AuNP (panel 4). To validate the amyloid formation on the AuNP, we cross-checked images obtained by HRTEM and cryo-TEM (Supplementary Fig. 6). In both the images, each AuNP was covered with an amyloid corona of uniform thickness (~3 nm), which corresponds to the size of a single oligomeric aggregate. This implies that the amyloid corona on the AuNP comprises a single layer of Aβ oligomeric aggregate[23]. We further investigated the role of Aβ oligomerization in the formation of the amyloid corona. To impede the Aβ oligomerization, we added the anti-aggregation agent rutin hydrate during PNAC synthesis (Supplementary Fig. 7). The results showed that the formation of the amyloid corona on AuNP was significantly restricted in the presence of the anti-aggregation agent. This is because rutin hydrate binds to monomeric Aβ and interferes with the formation of the amyloid corona on the PNAC. The results imply that a PNAC-based assay can be also utilized to discover the Aβ anti-aggregation compounds that block the surface-induced aggregation.

The formation of a sterically stable amyloid corona strongly depends on the initial amount of Aβ monomers (supplementary Fig. 8a). Specifically, for lower concentrations of Aβ monomers (<1.4 μM), the aggregation of PNACs is increased because the amyloid corona cannot fully cover the AuNP surface. The extent of PNAC aggregation is also increased at higher concentrations of Aβ monomers (>1.4 μM)[24]. It is therefore plausible that the Aβ monomer concentration is a major factor in Aβ aggregation on the PNAC. We determine that the optimal Aβ monomer concentration is 1.4 μM, at which the amyloid achieves full coverage of the AuNP surface with a uniform thickness (~3 nm), as seen in Fig. 1c. In this condition, the PNACs were successfully fabricated with a high yield (~95%) and good steric stability that was observed for several weeks (Supplementary Figs. 8 and 9). From the normalized absorbance spectra of the AuNPs and PNACs, the localized surface plasmon resonance (LSPR) peaks appeared at ~520 and ~525 nm, respectively (Supplementary Fig. 10). The peak shift of PNACs was reasonably attributed to the formation of the amyloid corona on the AuNPs. To monitor the amyloid degradation process on one AuNP over time, we performed cryo-TEM analysis of a single PNAC treated with Aβ-degrading agents (Fig. 1d). In the absence of Aβ-degrading agents, the amyloid corona remains intact with the approximately 3-nm thickness (Fig. 1d, panel 1). After the addition of the Aβ-degrading agent 4-(2-hydroxyethyl)-1-piperazine propanesulfonic acid (EPPS), the amyloid corona begins to decompose (Fig. 1d, panels 2 and 3) before disappearing (Fig. 1d, panel 4).

**Physical and chemical characterization of PNAC.** To characterize the conformational characteristics of the amyloid corona of the PNAC, we used graphene-based sensors[25] wherein each surface of the sensor was functionalized with three different types of antibodies (monoclonal 6E10, polyclonal A11, and polyclonal OC), as shown in Fig. 2a. Detailed information regarding these graphene-based sensors is provided in Supplementary Figs. 11 and 12. Because 6E10 is known to specifically recognize the sequence of Aβ$_{1-16}$, 6E10 can capture all conformational structures of Aβ, including Aβ monomers, oligomers, and fibrils[26]. For this reason, we used a 6E10-immobilized graphene sensor for a positive control experiment. A11 and OC antibodies specifically

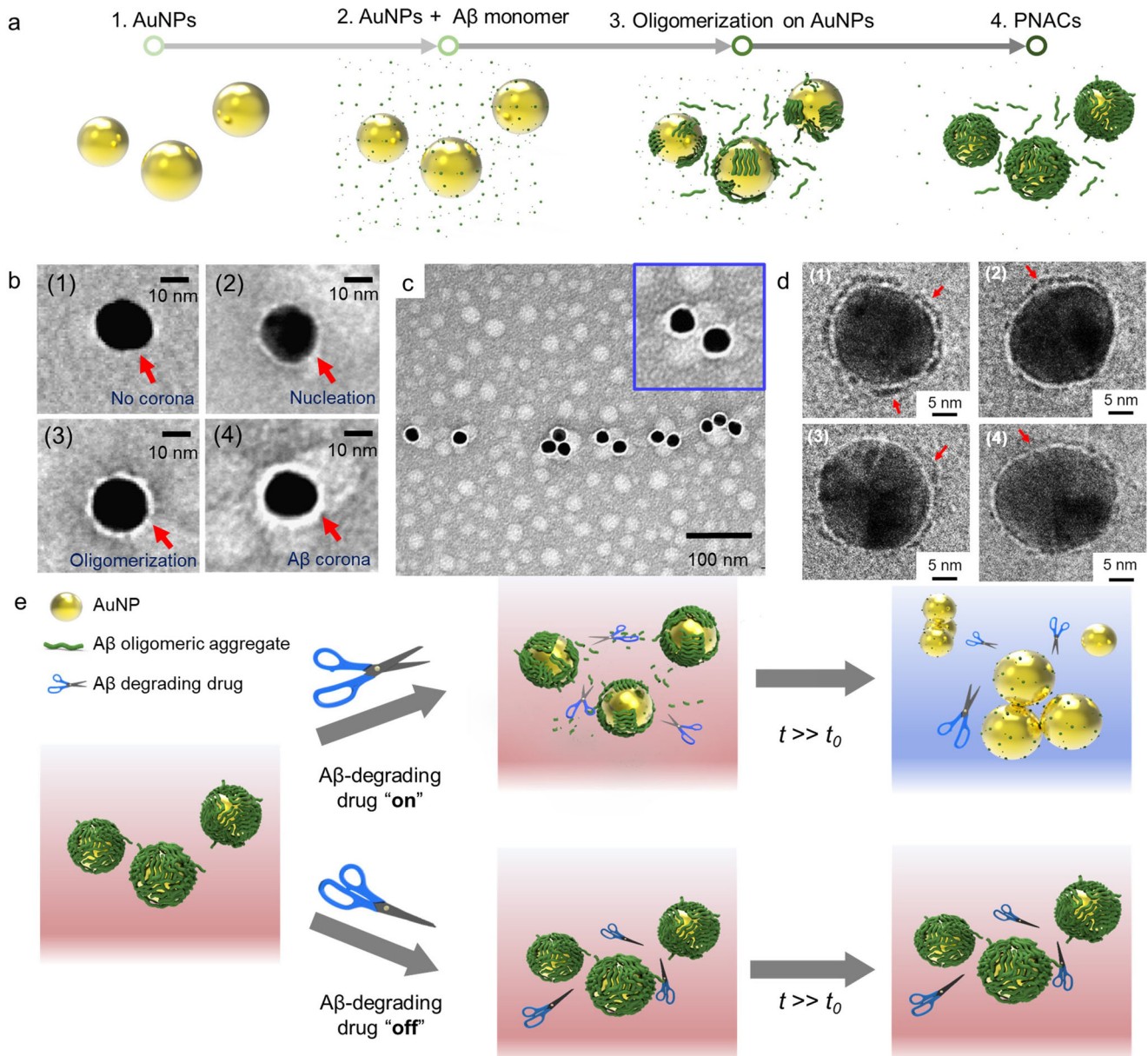

**Fig. 1 The fabrication of plasmonic nanoparticle amyloid corona (PNACs) and process of PNAC-mediated anti-Aβ drug screening. a** The formation process of PNACs using the catalytic property of gold nanoparticles (AuNPs). **b** HRTEM images of AuNPs (panel 1), intermediates (panels 2 and 3), and PNACs (panel 4). The red arrow represents the surface of the AuNP. **c** HRTEM images of PNACs. **d** Cryo-TEM images of the PNAC interacting with anti-Aβ drugs. Images from panels 1 to 4 represent the removal of the amyloid corona from the PNAC surface by anti-Aβ drugs. The red arrow represents the remaining amyloid corona on PNAC. **e** Schematic of performance of PNAC-mediated anti-Aβ drug screening platform.

bind to Aβ oligomers and Aβ fibrils, respectively. In principle, the electrical conductance of the graphene-based sensor is increased when PNACs are captured by the corresponding antibodies immobilized on the graphene film surface. This is because the base material of the PNAC is the AuNP, which can transfer free electrons to the graphene-based sensor, thus providing extra conduction paths. In cases with little PNAC–antibody affinity, the electrical conductance of the graphene-based sensor is unchanged.

Before the assay using PNACs, we checked the affinities between the antibodies and each of the purified Aβ species of purified Aβ monomers, oligomers, and fibrils (Supplementary Figs. 13–15). The results showed that each type of Aβ properly reacted with the corresponding antibody-immobilized graphene sensor (Supplementary Fig. 16). This confirmed that the antibody-immobilized

graphene sensor was suitable for determining the conformation of the amyloid corona on the PNACs. Then, we checked the affinities between the antibodies and bare AuNPs as a negative control. The relative resistance changes, which represent the affinity between bare AuNPs and the antibodies, are negligible from all antibodies (monoclonal 6E10, polyclonal A11, and polyclonal OC) (Fig. 2b). However, with the PNACs, the relative resistance values of the 6E10- and A11-immobilized sensors are significantly changed by 2.51 and 2.35%, respectively, implying that 6E10 and A11 antibodies have strong affinities with the PNACs. In contrast, the relative resistance value of the OC-immobilized sensor is unchanged, remaining similar to that of the sensor reacted with AuNPs, meaning that OC antibodies do not capture PNACs. These results indicate that the amyloid corona comprises oligomeric aggregates not Aβ fibrils. In addition, considering the similar

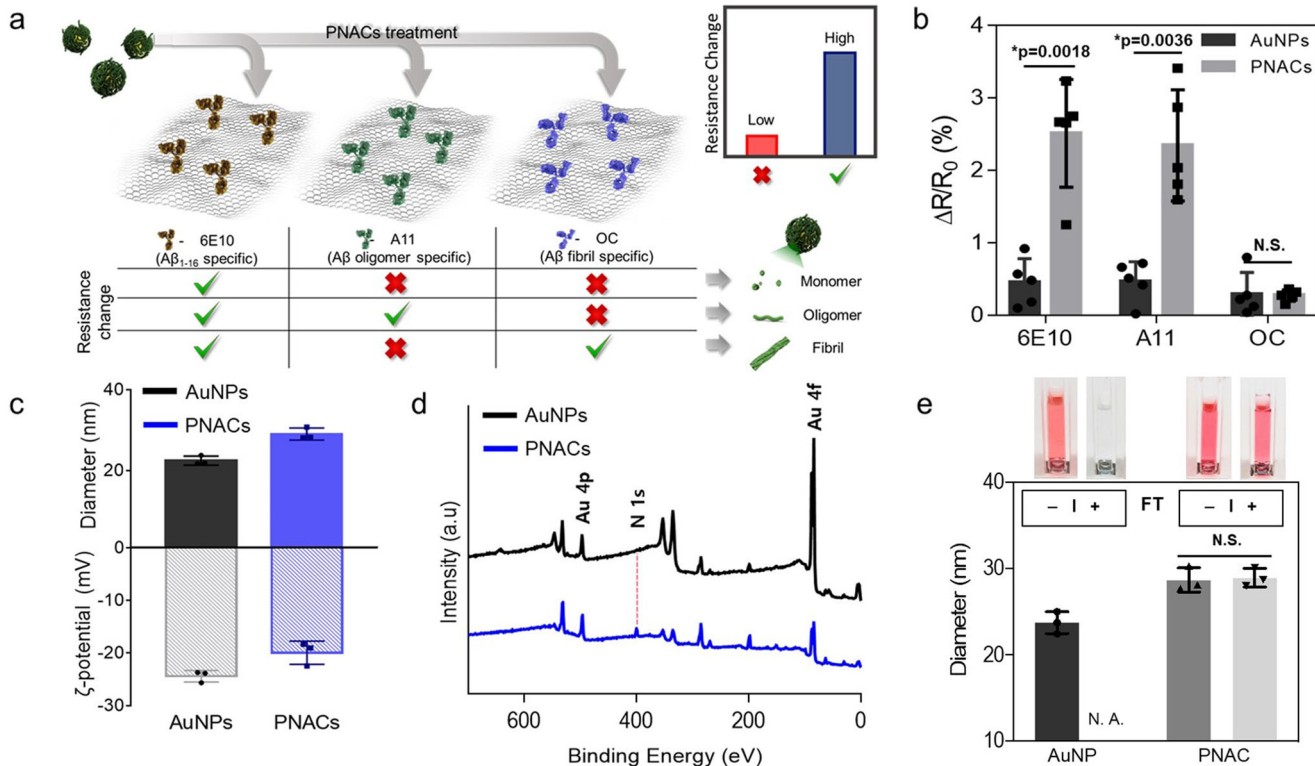

**Fig. 2 The physical and chemical characterization of PNACs. a** Schematic of the conformational characterization of amyloid corona using the graphene-based sensor with antibodies 6E10 (brown), A11 (green), and OC (blue). The red cross marks represent small changes in the resistance of the graphene-based sensor, whereas the green checkmarks represent big changes in the resistance of the graphene-based sensor. **b** The relative resistance changes $\Delta R/R_0$ of antibody-immobilized graphene sensors depending on their treatment with AuNPs (black bars) and PNACs (gray bars). Additional bars in **b** represent the average ± standard deviation calculated from $n = 5$ independent graphene-based sensors. **c** The hydrodynamic diameters and ζ-potentials of AuNPs and PNACs. **d** XPS survey spectra of AuNPs (black line) and PNACs (blue line). **e** Photographic images of the solution of AuNPs and PNACs before and after the freeze–thaw process and the changes in the hydrodynamic diameters of AuNPs and PNACs. FT freeze–thaw process, N.S. not significant, N.A. not available. Bar plots in **c** and **e** represent the average ± standard deviation calculated from $n = 3$ independent samples. Source data are provided as a Source Data file.

resistance changes shown by the 6E10- and A11-immobilized sensors, we conclude that the composition of the amyloid corona is not Aβ monomers, but Aβ oligomeric aggregates. Note that the polyclonal antibody A11 has been widely used to recognize a generic epitope in synthetic Aβ oligomers[20], but it does not bind low-molecular-weight oligomers (e.g., dimers and trimers), or protofibrils of double-cysteine mutant Aβ[8,27,28]. For this reason, it implicated that amyloid corona on PNACs consisted of A11-positive oligomeric aggregates[8] that were partially disordered due to their attachment onto the AuNP surface.

We attempted to check whether the growth of the amyloid corona on AuNPs saturated because the efficacy of oligomeric aggregate-targeting drugs could differ by the thickness of amyloid corona on AuNPs. To this end, we added Aβ monomers (5 or 10 μM) into a PNAC solution and checked the thickness of the amyloid corona on AuNPs (Supplementary Fig. 17). When a PNAC solution with 5-μM Aβ monomers was incubated for 2 days, we observed that the PNACs were attached to the newly formed Aβ fibrils (Supplementary Fig. 17a). Note that the thickness of amyloid corona on AuNPs does not change after the incubation with 10-μM Aβ monomers (Supplementary Fig. 17b). These results indicated that the conformation of the amyloid corona was stable even under the further addition of Aβ monomers, implying that the oligomeric aggregates on AuNP do not follow the pathway of amyloid fibril formation.

Physicochemical characterizations reveal that the PNACs are approximately 6 nm larger than the bare AuNPs (~22.45 nm) and

possess slightly reduced surface charges relative to bare AuNPs (Fig. 2c). These data are consistent with the HRTEM images showing the thickness (~3 nm) of the amyloid corona (Fig. 1c and Supplementary Fig. 6) and thus corroborate the uniformity of the PNACs. The results of X-ray photoelectron spectroscopy (XPS) also confirm that the amyloid corona encapsulates the AuNPs (Fig. 2d). In the PNAC spectrum, N 1s peaks (~400 eV) appear, and the intensity of the Au 4f peak is decreased compared to that of bare AuNPs, indicating that the amyloid corona covers the AuNP surfaces. Finally, we determined that the freeze–thaw stability of the PNACs was significantly improved compared to that of the AuNPs (Fig. 2e and Supplementary Fig. 18).

We confirmed that the size and dispersity of PNACs were fully recovered to their initial states after the freeze–thaw process. These results suggest that the PNACs have a good cryopreservation storage capacity that prevents Aβ denaturation. The results also indicate that the amyloid corona is a hard corona (i.e., or irreversibly bound to the AuNP surface)[29,30]. The stability of the PNAC is a necessary component of our proposed strategy for drug screening because the degradation of the amyloid corona must only be caused by oligomeric aggregate-degrading drugs.

**Kinetic analysis of Aβ-degrading proteases.** To validate our drug screening strategy, we first used protease XIV for Aβ decomposition as a positive control. Protease XIV is considered a digestive enzyme that can degrade all conformational structures

of Aβ, including oligomers and fibrils, into small peptide fragments[31]. To quantify the degradation of the amyloid corona with ultraviolet–visible (UV–vis) spectra, we adopted the relative absorbance ratio, as the absorbances at 609 and 525 nm ($A_{609}$ and $A_{525}$) denote the degrees of PNAC aggregation and dispersion, respectively (Supplementary Figs. 10 and 19). With protease XIV (100 μg mL$^{-1}$), the relative absorbance ratio ($A_{609}/A_{525}$) of the PNAC solution began to increase rapidly before saturating (see an inset graph in Supplementary Fig. 19). The LSPR peak of the PNACs is shifted strongly with variations in the protease XIV concentration, indicating that the amyloid corona is degraded by the activity of protease XIV (Fig. 3a). In contrast, no LSPR peak shift of the PNAC solution is observed with inactivated protease XIV (Fig. 3b).

From the UV–vis spectra, the $A_{609}/A_{525}$ ratio of the PNAC solution is found to increase with increasing protease XIV concentration (black bar in Fig. 3c), but no significant increase occurs with increasing inactivated protease XIV concentration (gray bar in Fig. 3c), thus confirming the activity of protease XIV. We also analyze the change in relative absorbance ($T_1 - T_0$) as a function of the protease XIV concentration using the dose–response model (Fig. 3d; see "Methods")[32]. We extract the values of the half-maximal effective concentration $EC_{50}$ and the maximal efficacy by the sigmoidal dose–response model (Eq. (1)).

$$\Delta \text{ Relative absorbance } (T_1 - T_0)$$
$$= \frac{1.023}{1 + 10^{\text{Log}29.83 - [\text{Protease XIV}]}}, R^2 = 0.93$$

(1)

The values of $EC_{50}$ and maximal efficacy are estimated as 29.83 μg mL$^{-1}$ and 1.023 (a.u.), respectively. Moreover, we performed time-dependent tests of protease XIV to investigate its time-variable effects (Supplementary Fig. 20a, b). In detail, we added 25, 50, 100, and 200 μg mL$^{-1}$ protease XIV to each PNAC solution to monitor the concentration- and time-dependent enzymatic activity. The results showed that the $A_{609}/A_{525}$ was increased with the concentration of protease XIV.

Next, we applied the PNAC platform to investigate the proteolytic activity of matrix metalloproteinase (MMP-9), which is one of the Aβ-degrading proteases used for the clearance of interstitial and endoplasmic reticulum/Golgi Aβ in the brain[33]. Reportedly, MMP-9 can degrade $A\beta_{1-42}$ fibrils but not $A\beta_{1-40}$ fibrils; however, the degradability of Aβ oligomers by MMP-9 remains an open question[34]. To answer this question, we demonstrated the PNAC platform as a tool for characterizing MMP-9 performance (Fig. 3e). We prepared two types of PNAC encapsulated with $A\beta_{1-40}$ or $A\beta_{1-42}$ oligomeric aggregates and apply MMP-9 at concentrations of 100 fg mL$^{-1}$ to 1 ng mL$^{-1}$ to these solution. While the amyloid corona comprising $A\beta_{1-40}$ oligomeric aggregates is not degraded by MMP-9, that comprising $A\beta_{1-42}$ is degraded by MMP-9 (Fig. 3f–i). These results are similar to a previous report[33] demonstrating that MMP-9 hardly degraded $A\beta_{1-40}$ fibrils but successfully degraded $A\beta_{1-42}$ fibrils. In particular, we also analyze the change in relative absorbance ($T_1 - T_0$) as a function of the MMP-9 concentration using the sigmoidal dose–response model. We extract the values of the half-maximal effective concentration ($EC_{50}$) and the maximal efficacy by the sigmoidal dose–response model, following (Eq. (2)) (Fig. 3j, k)

$$\Delta \text{ Relative absorbance } (T_1 - T_0)$$
$$= \frac{0.4213}{1 + 10^{\text{Log}1.283 - [\text{MMP}-9]}} + 0.03607, R^2 = 0.93$$

(2)

The values of $EC_{50}$ and maximal efficacy are estimated as 1.283 pg mL$^{-1}$ and 0.4213 (a.u.), respectively. In addition, we performed time-dependent tests of MMP-9 to investigate the time-variable enzymatic activity of MMP-9 (Supplementary Fig. 20c, d) with various MMP-9 concentrations of 10 fg mL$^{-1}$–10 ng mL$^{-1}$. The results show that the dose- and time-dependent proteolytic activity of Aβ-degrading protease (i.e., MMP-9) can be measured by our screening platform with a high analytical sensitivity of 100 fg mL$^{-1}$.

**Dose- and time-dependent tests for anti-Aβ agents**. Beyond studying the activity of Aβ-degrading proteases, we measured the efficacy of the anti-Aβ agents summarized in Table 1 using our screening platform. Before monitoring the efficacy of all anti-Aβ agents, a positive experiment was conducted with EPPS, which can degrade all Aβ species, including Aβ oligomers and fibrils, into monomeric Aβ[15]. From the UV–vis spectra of the PNAC solution in the presence of EPPS (Fig. 4a), the LSPR peak of the PNAC solution is shifted with increasing EPPS concentration (1–40 mM). The LSPR shift is significantly lower in the presence of EPPS than in that of Aβ-degrading proteases for the following reason. Theoretically, anti-Aβ drugs operate by three mechanisms: (i) solubilization by direct binding to Aβ, (ii) phagocytosis by activated microglia, and (iii) activation of Aβ extraction to the blood by the peripheral sink. Among these mechanisms, EPPS follows the solubilization mechanism by direct binding to Aβ[15]. EPPS dismantles the β-stacking inside Aβ oligomers and thereby cuts them into individual monomers[15]. However, protease XIV and MMP-9 can continuously degrade Aβ into numerous small peptide fragments by proteolysis until the termination of the enzyme–substrate reaction[31]. Therefore, Aβ degradation with the proteases occurs rapidly on the PNAC surface, which induces a more dramatic peak shift of the PNAC solution compared to that observed with EPPS.

The $A_{609}/A_{525}$ ratio of the PNAC solution is logarithmically increased as a function of the EPPS concentration, providing strong evidence of the ability of EPPS to degrade the amyloid corona of the PNAC (Fig. 4b). To investigate the pharmacokinetics of EPPS, we extract the values of the half-maximal effective concentration $EC_{50}$ and the maximal efficacy by the sigmoidal dose–response model (see the inset of Fig. 4b). The values of $EC_{50}$ and maximal efficacy are estimated as 7.691 mM and 0.7938 (a.u.), respectively, consistent with a previous report[15].

To investigate the time-variable effects of EPPS, we perform kinetic analysis of EPPS depending on degradation time up to 24 h with four different EPPS concentrations (Fig. 4c). As the EPPS concentration is increased, the time to reach 50% of Aβ-degrading drug activity (Time$_{50\%}$) and the time constant ($\tau$) is exponentially decreased (Fig. 4d) as shown by Eqs. (3) and (4), respectively.

$$\text{Time}_{50\%} = 2.472 \times e^{-0.0780[\text{EPPS}]} + 2.208$$

(3)

$$\text{Time constant } (\tau) = 3.567 \times e^{-0.0781[\text{EPPS}]} + 3.186$$

(4)

These results may help to optimize the binding kinetics of drug candidates and predict their time-dependent drug activities, which could facilitate the proper selection of the optimal concentrations of drug candidates for successful early clinical trials[35,36]. In addition, this approach can be extended to investigate the pharmacokinetic properties of other promising oligomeric aggregate-degrading drugs.

To further validate the versatility and practicality of our platform, we tested the efficacy of several anti-Aβ agents with known efficacy in treating AD. We measured the UV–vis spectra of PNAC solution in the presence of five different anti-Aβ agents

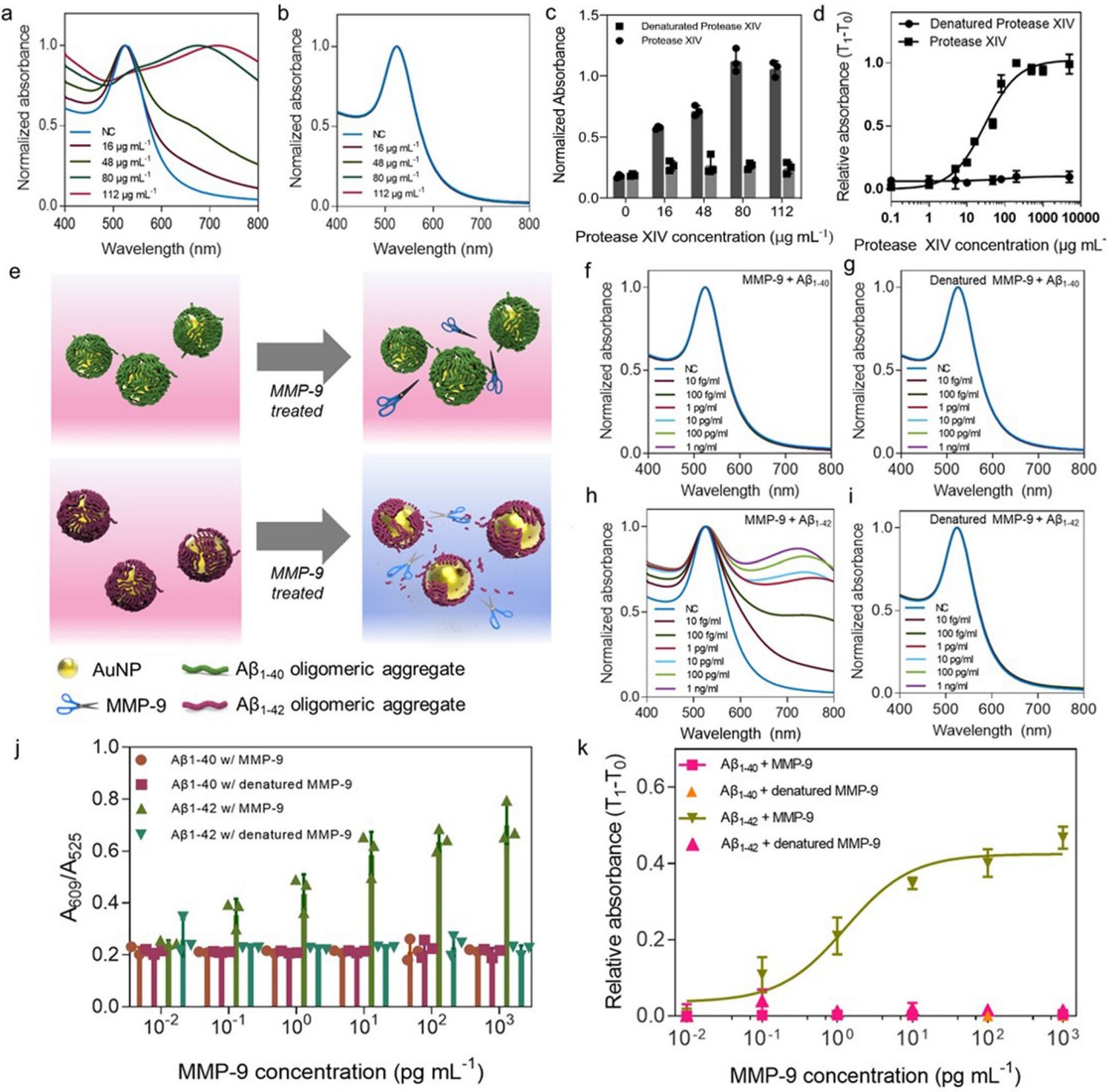

**Fig. 3 Kinetic analysis of Aβ-degrading enzymes using PNACs. a** UV–vis spectra of PNAC solution depending on active and denatured protease XIV concentrations of 0, 16, 48, 80, and 112 µg mL$^{-1}$. **b** UV–vis spectra of PNAC solution depending on the concentration of denatured protease XIV; the spectra are overlapped. **c** $A_{609}/A_{525}$ of PNAC solution depending on the concentration of protease XIV. **d** Plots fitted by the sigmoidal dose–response curve as a function of protease XIV concentration. **e** Schematic showing the colorimetric response of PNAC solution depending on the composition of the amyloid corona. **f–i** UV–vis spectra of PNACs encapsulated by Aβ$_{1-40}$ or Aβ$_{1-42}$ depending on the concentration of MMP-9 from 10 fg mL$^{-1}$ to 1 ng mL$^{-1}$. **f** PNACs encapsulated by Aβ$_{1-40}$ with MMP-9, where the spectra are overlapped. **g** PNACs encapsulated by Aβ$_{1-40}$ with denatured MMP-9, where the spectra are overlapped. **h** PNACs encapsulated by Aβ$_{1-42}$ with MMP-9. **i** PNAC encapsulated by Aβ$_{1-42}$ with denatured MMP-9, where the spectra are overlapped. **j** The $A_{609}/A_{525}$ of each PNAC solution acquired from (**f–i**). **k** The plot fitted by the sigmoidal dose–response curve as a function of MMP-9 concentration from 10 fg mL$^{-1}$ to 1 ng mL$^{-1}$. Bar and dot plots in **c**, **d**, **j**, and **k** represent the average ± standard deviation calculated from $n = 3$ independent samples. Source data are provided as a Source Data file.

of curcumin, glutathione, rutin hydrate, a polyphenol (−)-epi-gallocatechin gallate (EGCG), and tramiprosate (Fig. 5a–e). Detailed information on these agents, including their chemical names, molecular structures, critical roles in Aβ inhibition or degradation, and concentration ranges, is listed in Table 1. The LSPR peak of each PNAC solution treated with curcumin, EGCG, and rutin hydrate is changed slightly, while that of the PNAC solution with glutathione and tramiprosate is unchanged.

Considering the role of each tested anti-Aβ agent, these results are plausible. Reportedly, glutathione inhibits neuroinflammation in the brain; this has no connection to the degradation of oligomeric aggregate[37]. Similarly, tramiprosate and rutin hydrate have only slight effects on the degradation of oligomeric aggregate, because these agents only inhibit Aβ oligomerization[38,39]. In contrast, curcumin, known to disassemble amyloid plaques, slightly affects Aβ oligomer degradation[40]. EGCG affects the

**Table 1 List of anti-Aβ drug candidates used for the validation of the PNAC-based colorimetric drug screening platform. The structure of protease XIV (PDB ID:1OS8) and active site of MMP-9 (PDB ID:1L6J) are shown as cartoon representation.**

| Category | Name & molecular weight | Molecular structure | Critical role in Aβ inhibition or degradation | Concentration range in this study | Reference |
|---|---|---|---|---|---|
| Protein | Protease XIV (23 kDa) | | Very fast Aβ degradation into small fragment peptide | 16–5000 µg mL$^{-1}$ | 31 |
| | Matrix Metalloproteinase-9 (MMP-9, 82 kDa) | | Clearance of interstitial and ER/Golgi Aβ | 100 fg mL$^{-1}$–1 ng mL$^{-1}$ | 34 |
| Small molecule | 4-(2-hydroxyethyl)-1-piperazinepropanesulphonic acid (EPPS, 252 Da) | | Degradation from conformational Aβ species to Aβ monomer | 0–40 mM | 15 |
| | Glutathione (307 Da) | | Reduced the oxidative stress and neurodegeneration | 0–10 mM | 37 |
| | Tramiprosate (139 Da) | | Binding to Aβ peptides and inhibiting Aβ fibrilization | 0–10 mM | 38 |
| | Curcumin (368 Da) | | Inhibiting amyloid formation, oxidant and reducing amyloid plaque | 0–10 mM | 40 |
| | Rutin hydrate (628 Da) | | Ameliorating oxidative stress and inhibiting of Aβ oligomerization | 0–10 mM | 39 |
| | Polyphenol (−)-epigallocatechin gallate (EGCG, 458 Da) | | Degrading Aβ fibrils into smaller, amorphous protein aggregates | 0–10 mM | 41 |

degradation of oligomeric aggregate as much as curcumin does because it can remodel Aβ oligomers and fibrils into nontoxic aggregates by binding to the β-sheets of Aβ species[41]. The results reveal that all mentioned anti-Aβ agents have much less influence on the degradation of oligomeric aggregate than EPPS has.

To extract the pharmacokinetic parameters of these anti-Aβ agents (i.e., correlation coefficient $R^2$, hillslope, EC$_{50}$, and maximal efficacy), we analyze the A$_{609}$/A$_{525}$ ratio of each PNAC solution depending on the concentration of anti-Aβ agents by dose-dependent curves (Fig. 5f, g). For glutathione and tramiprosate, no parameters are extracted because these agents are ineffective in oligomeric aggregates on PNAC. The hillslope of rutin hydrate is the steepest among the anti-Aβ agents, while each anti-Aβ agent shows similar potency, as judged by the calculated EC$_{50}$. Meanwhile, the maximal efficacy of EPPS is the highest among the anti-Aβ agents. To better understand which anti-Aβ agent is more effective, we introduce an index defined as the value

of maximal efficacy divided by the value of EC$_{50}$. By this index, EPPS is at least 2.5 times more effective than the other anti-Aβ agents (Fig. 5h).

To investigate whether PNAC-based drug screening can function in biological conditions, we synthesized PNACs within the medium of human cerebral spinal fluid (hCSF). We validated the formation of the amyloid corona on hCSF-PNACs by HRTEM, the graphene-based sensors, and gel electrophoresis (Supplementary Fig. 21), as done for the PNACs synthesized in PBS. The results indicate that the amyloid corona of hCSF-PNACs also comprises A11-positive oligomeric aggregates. Moreover, to verify the robustness of our drug screening platform, we tested the dose–response curves of protease XIV and EPPS with the hCSF-PNACs (Supplementary Figs. 22 and 23). The results are exactly consistent with those of PNACs in PBS. Altogether, the results demonstrated the potential ability of the PNAC-based platform to facilitate the discovery of promising Aβ oligomeric aggregate-degrading agents.

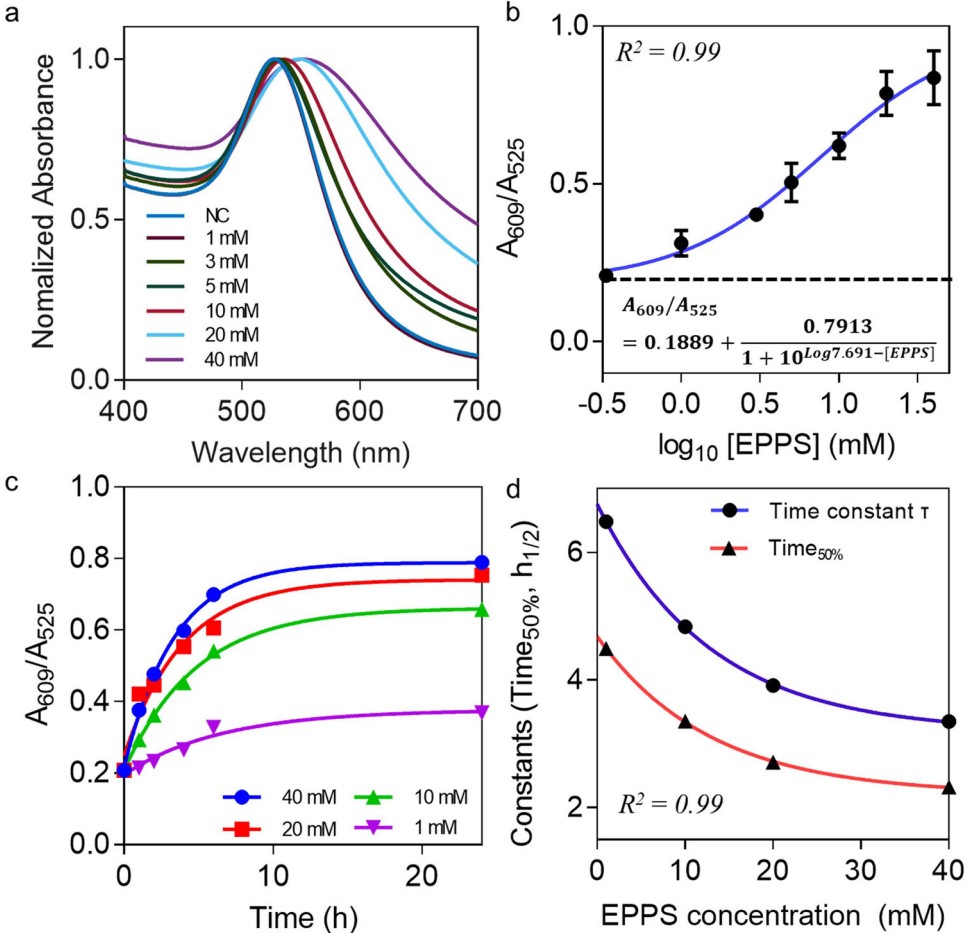

**Fig. 4 Kinetic analysis of Aβ-degrading agent (EPPS). a** UV–vis spectra of PNAC solution depending on EPPS concentration (0, 1, 3, 5, 10, 20, and 40 mM). **b** $A_{609}/A_{525}$ of PNAC solution depending on the concentration of EPPS. **c** $A_{609}/A_{525}$ of PNAC solution depending on the reaction time (1–24 h) with various concentrations of EPPS. **d** The kinetic analysis of the degradation of oligomeric aggregate depending on the EPPS concentration. Dot plots in **b** represent the average ± standard deviation calculated from $n = 3$ independent samples. Source data are provided as a Source Data file.

In summary, we have described a strategy for the rapid screening of Aβ oligomeric aggregate-degrading agents based on the colorimetric response of a PNAC solution. Our obtained results suggest that this approach is highly effective for the discovery of drugs that degrade oligomeric aggregates, which are currently the main target molecules considered in the treatment of AD. Because the PNAC-based strategy relies on the optical properties of a plasmonic nanoparticle encapsulated with a uniform oligomeric aggregate-based hard corona, it can provide a precise and quantitative description of the degradation of the oligomeric aggregate without the use of chemical stains or antibodies. The ability of our platform to quantify the degradation of oligomeric aggregate was validated via the kinetic analysis of protease XIV and MMP-9. In addition, the efficacy of other small molecules in degrading oligomeric aggregates was monitored through our platform, and the colorimetric responses of the PNAC solution were analyzed through both time- and dose-dependent pharmacokinetics. The results of the analysis confirmed the capacity of the system for clear comparisons of efficacy among anti-Aβ agents. Considering these findings, we anticipate that the PNAC-based platform will allow the screening of chemical databases for the identification of pools of potent small molecules against Aβ oligomeric aggregates. This approach will also contribute to investigating the pharmacokinetic properties of oligomeric aggregate-degrading agents and may be used to find compounds that block surface-induced aggregation.

## Methods

**Reagents**. Aβ$_{1-40}$ and Aβ$_{1-42}$ were purchased from Tocris Co. (UK) with >95% purity. Chloroauric acid trihydrate (HAuCl$_4$·3H$_2$O), trisodium citrate, EPPS, glutathione, tramiprosate, EGCG, rutin hydrate, and curcumin were purchased from Sigma-Aldrich (USA). Protease XIV and MMP-9 were purchased from Thermo Fisher Scientific (USA). Monoclonal 6E10 antibodies (purified anti-β-amyloid antibody, 1–16, BioLegend, San Diego, CA, USA), polyclonal A11 antibodies (AHB0052, Invitrogen, USA), and polyclonal OC antibodies (200-401-E87, Limerick, PA, USA) were also purchased.

**Synthesis of AuNPs**. All glassware was soaked in aqua regia (a mixture of HNO$_3$ and HCl at a 1:4 ratio) and rinsed with deionized water (DW) followed by Millipore water before fabricating AuNPs. The AuNPs were synthesized by the citrate reduction method. In brief, to prepare AuNPs with a 20-nm diameter, 2.55 ml of 38.8-mM HAuCl$_4$ solution (HT-1004, Sigma-Aldrich, USA) was mixed with the 47.5 mL of Millipore water, and the mixed solution was heated while vigorously stirring (~1400 rpm). When the solution began to reflux, 0.9 mL of 38.8-mM sodium citrate was added quickly. The solution was allowed to reflux for 30 min and then cooled naturally to room temperature (20–25 °C) under vigorous stirring (~1400 rpm).

**Preparing the Aβ monomers in vitro**. To obtain monomeric Aβ, lyophilized Aβ peptides (Tocris Co. UK) were dissolved in 1,1,1,3,3,3-hexafluoro-2-propanol (HFIP, Sigma-Aldrich). The obtained Aβ solution (1 mg mL$^{-1}$) was incubated at room temperature for 1 h. The Aβ solution was distributed into 1.7-mL microcentrifuge tubes and then dried overnight in a fume hood. The microcentrifuge tubes were transferred to a SpeedVac concentrator (Labogene, Denmark) and centrifuged under vacuum at −20 °C for 1 h to remove any remaining traces of HFIP and moisture. The formed Aβ monomer solution (10 µM) was stored at −20 °C as a stock solution. Before the synthesis of PNAC, the Aβ monomer solution was diluted with DI water and incubated at 37 °C for 1 h.

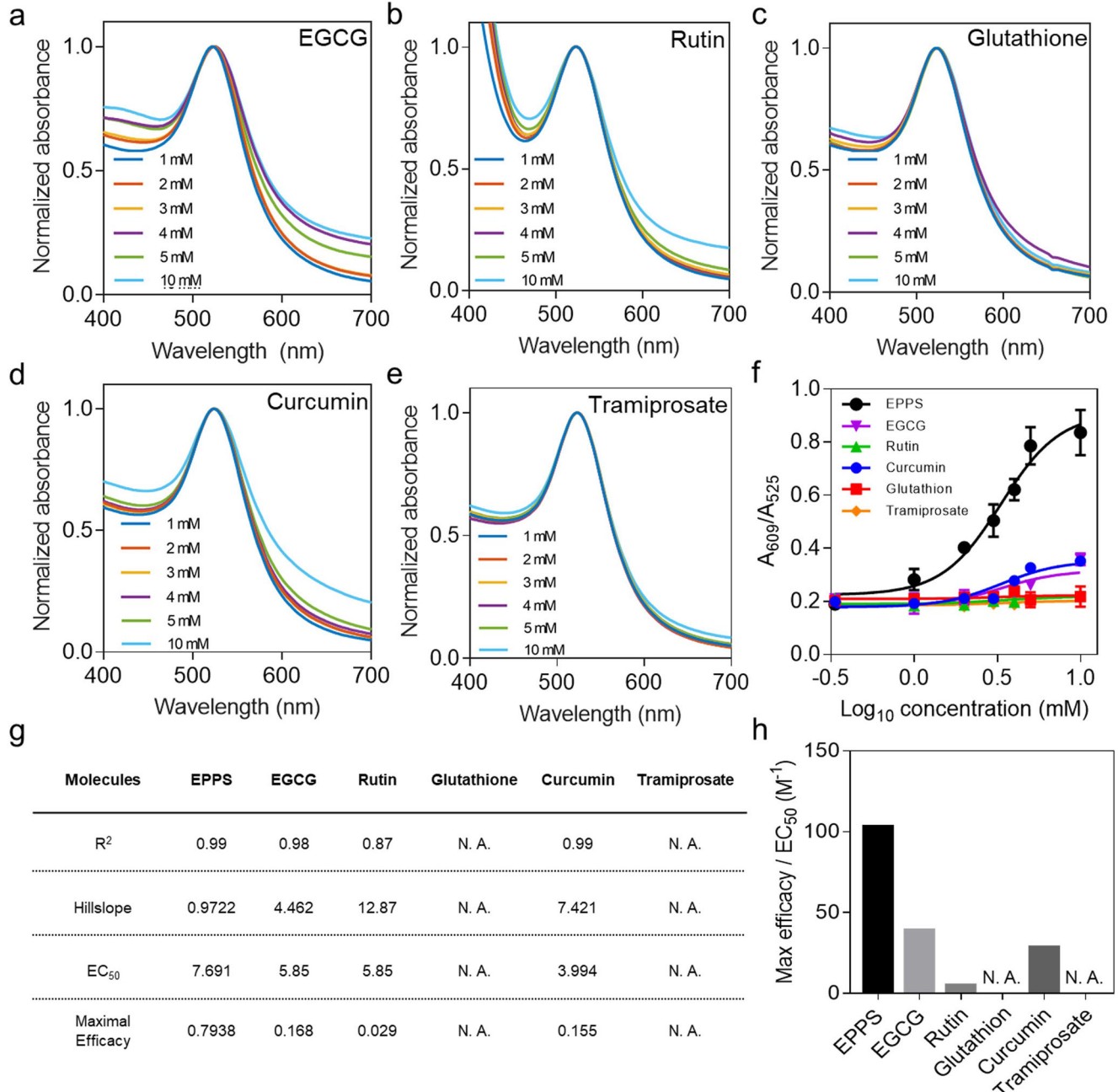

**Fig. 5 Monitoring the Aβ-degrading efficacy of anti-Aβ drug candidates. a–e** UV–vis spectra of PNAC solution depending on various concentrations (1, 2, 3, 4, 5, and 10 mM) of each anti-Aβ drug candidate (i.e., EGCG, rutin hydrate, glutathione, curcumin, and tramiprosate). **f** The dose-dependent curve for $A_{609}/A_{525}$ of the PNAC solution treated with each anti-Aβ drug candidate. **g** A table summarizing the coefficient of correlation, hillslope, $EC_{50}$, and maximal efficacy for each anti-Aβ drug candidate (N.A. not available). **h** The effectiveness of each anti-Aβ drug candidate analyzed by the maximal efficacy normalized by $EC_{50}$. Dot plots in **f** represent the average ± standard deviation calculated from $n = 3$ independent samples. Source data are provided as a Source Data file.

**Preparation of PNAC**. To adjust the pH of the Aβ solution, 0.1× PBS with pH 7.4 was added to the Aβ solution. The Aβ monomeric solution was mixed with concentrated AuNP solution to reach final concentrations of 0.1–10 μM. In detail, we removed the AuNP solution supernatant by centrifugation (6720*g*, 15 min) before adding 70 μL of the final AuNP solution to 160 μL of the Aβ monomer solution. The mixed solution was incubated for 24 h in a thermomixer (Eppendorf, Germany) with controlled incubation conditions (37 °C with 120-Hz shaking) to fabricate the PNAC solution. The final solution was changed to a suitable buffer solution (PBS or Tris buffer) for different detection conditions, and the final PNAC yield was confirmed by HRTEM imaging and the hydrodynamic diameters (Zetasizer Nano S90, Malvern instrument, UK). We used 2% uranyl acetate as a negative-staining solution in HRTEM imaging to enhance the contrast of the amyloid corona of the PNACs.

**The fabrication process of the graphene-based device**. Reduced graphene oxide (rGO) devices were fabricated via standard microfabrication techniques in a clean room (Supplementary Fig. 12). SiO₂ substrates of 300 nm in thickness were thermally deposited on 4-in. Si wafers and cleaned with a piranha solution (H₂SO₄:H₂O₂ = 4:1). These substrates were then dipped in 5% 3-(ethoxydimethylsilyl)propylamine (APTES, 97%, Sigma-Aldrich) solution diluted in ethanol for 2 h to induce amine functional groups (–NH₂) on the substrate surfaces. The amine functional groups were intended to improve the adhesion between the graphene oxide (GO) flakes and the SiO₂ substrates. The GO solution was obtained using the modified Hummer's method and dispersed in DW. The GO solution was spin-coated on the APTES-treated SiO₂ substrate at 3000 rpm for 60 s to obtain a GO thin film. Finally, rGO thin films were obtained by chemical reduction using hydroiodic acid (HI) at 80 °C for 3 h and patterned via conventional photolithography and reactive-ion etching with

patterns of 40 μm × 80 μm (width × length) in size. For the electrode fabrication, Au (150 nm) and Cr (30 nm) were deposited on the rGO-patterned SiO$_2$ wafer by an electron-beam (E-beam) evaporator. The Cr/Au layer was patterned via a lift-off process. To prevent the effect of nonspecific binding, the Au electrodes were passivated by photoresist (SU-8 2005, MicroChem), which is a well-known biocompatible and robust material. Consequently, the rGO patterns were partially exposed as binding sites for the antibodies.

**Functionalization of antibodies on the graphene-based sensor.** For antibody immobilization, the rGO-patterned devices were immersed in 50 mM 1-pyrenebutyric acid N-hydroxysuccinimide ester (PBSE, Sigma-Aldrich) diluted in dimethyl sulfoxide (DMSO) for 2 h at room temperature. The PBSE contained pyrene functional groups that enabled irreversible adsorption onto the rGO flakes via π–π interactions. Then, the devices were sequentially rinsed with DMSO and DW. Following PBSE functionalization of the rGO surface, 10 μg mL$^{-1}$ solution of antibodies in 0.1× PBS (pH 7.4) were incubated for 1 h to facilitate covalent bonding between the antibodies and the PBSE on the rGO surface. After the antibody immobilization, the devices were sequentially rinsed with 0.1× PBS and DW. Subsequently, the devices were incubated with 20 μL of the bare AuNPs and PNACs for 30 min to investigate the conformational composition of the PNAC. After the reaction, the devices were sequentially rinsed with 500 μL of 10 mM PBS and DW to remove the nonspecific binding materials from the device surfaces. The relative resistance values of each rGO sensor were measured both before the reaction ($R_b$) and after the reaction ($R_a$) between the antibodies and the nanoparticles (i.e., AuNPs and PNACs). The signal difference, ΔR, was defined as the absolute value of $\Delta R = R_a - R_b$, which demonstrated the reactivity between the antibodies and the nanoparticles. The electrical properties of the rGO devices ($n = 5$) were measured with a semiconductor parameter analyzer (KEITHLEY 4200A-SCS, USA). For comparison of the two groups, a one-tailed unpaired $t$ test was used, implemented by GraphPad Prism 7 software.

**Freeze–thaw process of PNAC and AuNP solution.** We confirmed the nature of the amyloid corona of the PNACs using a freeze–thaw process (Supplementary Fig. 18). When the solution of PNACs and AuNPs were frozen in a deep freezer at −80 °C for 1 h, the colors of the solution were changed to gray and light purple, respectively. Each solution was then defrosted at room temperature for 3 h. The color of the AuNP solution was unchanged after thawing, whereas the color of the PNAC solution was restored to wine red. The results indicated that the PNACs remained well dispersed. The hydrodynamic diameter was also unchanged from the initial condition (28.82 ± 1.02 nm) after freeze-thawing (28.73 ± 1.12 nm).

**XPS analysis.** All samples (AuNPs and PNACs) for spectral analysis were prepared from pellets after two centrifugation cycles in ultrapure water. After most of the supernatant was removed, the pellet (<70 μL) was spotted onto a Si substrate. XPS measurements were performed on a K-alpha instrument (Thermo VG, UK) using a monochromatic X-ray source (Al Kα line: 1486.6 eV) at $4.8 \times 10^{-9}$ mb. All samples were run as insulators using a low-energy flood gun for charge compensation. A survey scan of the 0–700 eV binding energy range and elemental scans of Au 4p, N 1s, and Au 4f were acquired from four spots on each sample, using a pass energy of 40 eV.

**UV–vis absorbance measurement.** The UV–vis absorption spectra (400–800 nm in wavelength) of the PNAC solution were recorded spectrophotometrically (Perkin Elmer, USA) at a rate of 600 nm min$^{-1}$. To quantify the LSPR shift, we adopted the relative absorbance ratio ($A_{609}/A_{525}$), which indicates the aggregation of PNAC because the absorbances at 690 and 525 nm ($A_{609}$ and $A_{525}$) denote the degrees of aggregation and dispersion, respectively. Photographs recording the color changes of the solution were taken with an iPhone XR. The TEM samples were stained using uranyl acetate to dye the Aβ proteins on the surfaces of the AuNPs.

**Kinetic analysis of protease XIV.** Before the kinetic analysis of protease XIV, we dissolved protease XIV in 1× PBS. To completely dissolve various concentrations of protease XIV (16–5000 μg mL$^{-1}$), we incubated the protease XIV solution at room temperature for 30 min. The kinetic analysis was performed with various concentrations of protease XIV added to each PNAC solution (total volume: 1 mL) and incubated for 1 h at 37 °C with low-speed shaking (120 rpm). For the negative control experiment, protease XIV was inactivated by incubation at 90 °C for 2 h. For a precise sigmoidal dose–response fitting, we defined the change in relative absorbance ($T_1 - T_0$) as the $A_{609}/A_{525}$ difference between before ($T_0$) and after protease treatment ($T_1$), such as protease XIV or MMP-9. The change in relative absorbance of each solution (PNAC solution treated with active or inactive protease XIV) was fitted to a sigmoidal dose–response curve to analyze the enzymatic activity of protease XIV.

**Kinetic analysis of MMP-9.** To investigate the Aβ oligomeric aggregate-degrading ability of MMP-9, we dissolved MMP-9 in 50 mM HEPES buffer (pH 7.5) with 10 mM CaCl$_2$ for 30 min. The kinetic analysis was performed by adding various concentrations (100 fg mL$^{-1}$–1 ng mL$^{-1}$) of MMP-9 to each PNAC solution (total

volume: 1 mL) and incubating for 3 h at 37 °C with low-speed shaking (120 rpm). To perform the negative control experiment, MMP-9 was inactivated by incubation at 90 °C for 2 h. The change in relative absorbance ($T_1 - T_0$) of each solution (PNAC solution treated with active or inactive MMP-9) was fitted to a sigmoidal dose–response curve to analyze the enzymatic activity of MMP-9.

**Gel electrophoresis of the PNAC with Aβ-degrading agents.** Agarose gel electrophoresis of PNAC was performed in the presence of different Aβ-degrading agents (i.e., EPPS and protease XIV). Each PNAC solution with the Aβ-degrading agent was loaded into a separate well of a 1.0 wt% agarose gel (Type I–A, low EEO). The electrophoresis was conducted at 100 V for 1 h.

**Cryo-TEM image analysis of the PNACs.** The PNAC solution was centrifuged at 6,720 g for 15 min to remove the supernatant of the PNAC solution. The solution was resuspended in DW and loaded onto copper grids. The samples were negatively stained with 2% uranyl acetate and quench-frozen into liquid ethane using a gravity-driven plunger apparatus. The samples were imaged at −170 °C in liquid nitrogen and observed by field-emission gun transmission electron microscopy operated at 200 kV (Tecnai F20, FEI).

**Separation of Aβ oligomer using sucrose gradients.** Density-gradient centrifugation was performed in the absence of agitation with the Aβ solution, which was incubated for 24 h at 4 °C. The 3 mL of Aβ solution were gently injected into a sucrose step gradient from 10 to 40% with a 10% increment. To separate the Aβ aggregates depending on the conformation and density, the sucrose solutions were centrifuged for 12 h at 23,000g in an SW 70 Ti swinging-bucket rotor (Beckmann Coulter, Fullerton, CA, USA) at room temperature (25 °C). The fractions were collected immediately and stored separately at 4 °C before the sampling for AFM images.

**Preparation of Aβ fibril.** To synthesize Aβ fibrils, an aliquot of lyophilized Aβ peptides was resuspended in a 10-mM HCl solution to adjust pH 2–3. By controlling the volume of the solution, we adjusted the final concentration of the solution as 100 μM. The solution was incubated in a thermomixer (Eppendorf, Germany) at 37 °C for 3 days. To remove extra Aβ monomers, the solution was then dialyzed against DW for 48 h by using 50-kDa dialysis membranes (Biovision Inc., USA).

**Atomic force microscopy (AFM) imaging of Aβ species.** To study the topological conformation and height of various Aβ species, 100 μL aliquots of each Aβ sample were deposited on the freshly cleaved mica substrate at room temperature (25 °C) for 30 min, rinsed with 200-μL DW, and then dried under a gentle flow of nitrogen gas to avoid physical aggregation of Aβ species before tapping-mode AFM imaging was performed. AFM images were generated by an NX10 (Park Systems, South Korea) equipped with a silicon tip (NCHR, Park Systems, South Korea) with a nominal radius of <10 nm. Image flattening and single-aggregate statistical analysis were performed by Smart Scan (Park Systems, South Korea). All tapping-mode images were produced with an image size of 10 μm × 10 μm at a scanning rate of 0.5-Hz.

**Preparation of PNACs in the cerebral spinal fluid (CSF).** To mimic Aβ oligomers generated in the human brain, we incubated Aβ solution in human CSF (hCSF, Lee Biotech, USA) for 24 h at 4 °C. The Aβ oligomers were mixed with the concentrated AuNPs to the final concentration of 1.4 μM before incubating for 24 h in a thermomixer (Eppendorf, Germany) with controlled incubation conditions of 37 °C with 120-Hz shaking to fabricate the hCSF-PNAC solution. The final solution was changed to a suitable buffer solution of PBS or hCSF for different detection conditions, and the formation of the final hCSF-PNACs was confirmed by HRTEM imaging and electrophoresis with a 1.0 wt% agarose gel.

**Statistics and reproducibility.** All values with an error bar were expressed as average ± standard deviation. Statistically significant differences were assessed by unpaired two-tailed Student's $t$ test for comparison between two groups. A $P$ value <0.05 was considered statistically significant. All results are from at least three independent experiments.

**Reporting summary.** Further information on research design is available in the Nature Research Reporting Summary linked to this article.

## Data availability
All data supporting the findings of this study are available within the article and its Supplementary Information files or from the corresponding author upon reasonable request. Source data are provided with this paper. The Source data file includes raw data underlying the respective main text (Figs. 1–5) and Supplementary Information (Supplementary Figs. 3, 5, 7–10, 16, 19, 20, 22, and 23). A reporting summary for this paper is available as a Supplementary Information file.

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

## Acknowledgements

This work was supported by the National Research Foundation of Korea (NRF) Grant funded by the Korean Government (MSIP) (No. NRF-2018M3C1B7020722, NRF-2019R1A2B5B01070617, and NRF-2020R1A2C2102262). This study was also supported by the Brain Korea 21 (BK21 FOUR). This research was also supported by the Health Fellowship Foundation and the Korea University Graduate School Junior Fellow Research Grant. G.L. gratefully acknowledges the financial support from the Korea University Grant. J.H. Lee was supported by a research grant from Kwangwoon University in 2021.

## Author contributions

D.L., J.H.L., G.L., and D.S.Y. conceived the study and wrote the paper, which was reviewed by all authors. D.L., D.P., I.K., S.W.L., and W.L. performed the experiment. D.L., D.P., K.S.H., G.L., and D.S.Y. analyzed the data. All authors agreed on the presentation of the paper.

## Competing interests

The authors declare no competing interests.
