## [Peer Review File · Nature Communications]

Reviewer #1 (Remarks to the Author):

This manuscript describes development of a plasmonic nanoparticles platform for monitoring the effect of drugs on degradation of toxic amyloid beta (A β) oligomers. The conducted experiments and the outcomes almost validated the authors' claims about their approach. There are several comments that needs to be properly addressed by the authors to improve the robustness of this approach:

- 1- Any suitable assays that monitors the formation and degradation of A β oligomers should be benign to the oligomerization and degradation processes. How the authors ensure that the degradation process is not affected by the plasmonic nanoparticles? What about the role of nanoparticles on the drugs and their degradation capacities? How the surface curvature of nanoparticles can affect the degradation process (as surface curvature can significantly affect biomolecule-nanoparticle interactions)?**
- 2- To ensure the robustness of this approach, the authors are encouraged to reproduce their outcomes on various sizes of the plasmonic nanoparticles.**
- 3- The multi later corona formation may critically interfere with the degradation process. The authors may perform cryo-TEM and electron tomography to better interpret the formation of amyloid corona at the surface of nanoparticles.**

Reviewer #2 (Remarks to the Author):

In this manuscript, authors describe a method called plasmonic nanoparticle amyloid corona (PNAC) to degrade the toxic amyloid β (A β) oligomers, which is a central feature of the onset and progression of Alzheimer's disease (AD), and monitor the drug-induced degradation of A β oligomers by analyzing the colorimetric responses of PNACs. The authors used A β -degrading proteases (protease XIV and MMP-9) and subsequently various small-molecule substances that have shown benefits in the treatment of AD. Although the drug screening technique has been shown to be meaningful in terms of specificity using plasmonic resonance peak shift, it is necessary to reinforce the interpretation from the scientific and clinical point of view, rather than a simple listing of various drug test results. To be published in this journal, clearer logical and authentic take-home messages are missing, and authors need to address following issues regardless of publishing another journal:

Major issues:

- 1. In p3, the authors wrote "First, we synthesized PNACs with AuNPs (20.1 \pm 1.2 nm)". Why the size of AuNPs were chosen to 20 nm? It would be better if authors wrote interpretation of choosing this size of AuNPs. For example, this size was chosen because it is (1) the size of plasmonic peak resonance which can effectively differentiate absorbance peak of AuNPs and PNACs or (2) low aggregation condition or etc.**
- 2. In this paper, "the relative absorbance ratio (A609 / A525)" is used a lot. It is necessary to mention what the absorbance values of 609 nm and 525 nm mean.**

3. Although the optimal A β oligomer concentration was found in Fig. S2a, AuNP aggregation seems to be occurred about 50 % in Fig. 1d. However, only 5 nm peak shift occurred in Fig. S3. In the background of Fig. 1d, a significant amount of A β fibrin particles estimated to be similar to the size of AuNPs were observed. Did this not affect the absorbance spectrum?

4. In p5, the authors wrote “Physicochemical characterizations reveal that the PNACs are approximately 6 nm larger than the bare AuNPs (~22.45 nm) and possess slightly higher surface charges than bare AuNPs (Fig. 2c).”. However, since the absolute value of the zeta potential of PNAC is smaller than that of the Au bare NP, the expression of high surface charge is incorrect.

5. In this paper, only antibody tests were performed using graphene-based sensors, and no data on electrical conductance measurements related to drug screening. It is not necessary that the sensor was fabricated using graphene rather than general-purpose Si in order to find antibodies to immobilize the a β oligomer. If graphene is used, electrical conductance measurement data related to drug screening should be added to Fig. 6 in order to increase sensitivity than conventional sensors. It is recommended to send the contents of Fig 2 as supplementary as it may interfere with readers' understanding of the paper.

6. Why did dose- and time-dependent tests run on small molecules (EPPS), not on proteins (Protease XIV, MMP-9)?

7. In Fig. 4d, the authors introduced Half-life and Time constant values for prediction of appropriated drug treatment time as mentioned in p8 “These results help to predict the appropriate treatment time depending on the EPPS concentration.”. However, the specific clinical usage of these values was not written, and it is difficult to understand the difference between them.

8. In Fig. 3a and 3h using high doses of protein (Protease XIV, MMP-9), the absorbance peaks changed significantly, indicating that the AuNP array was broken. However, in Fig. 4a using small molecules (EPPS), only small peak shifts due to degradation are observed. The authors should provide a concrete interpretation from a scientific point of view and Alzheimer's disease (AD) treatment point of view.

Minor issues:

1. In Fig.1b, 1c (inset), 1d (inset), scale bars are missing.

2. In Fig. S2b and S2c, the authors should write the legend of DI and PBS.

3. In Fig. 3b, the authors should clearly indicate on the graph or caption whether cases (1)-(5) were overlapped.

4. In Fig. 3a, 3f-3l, 4a, 5a-5e, throughout the graph, the authors did not use common line colors, and the direction of the arrows, not marked with a clear legend, can cause confusion for the readers.

Response Letter

Reviewer #1 (Remarks to the Author):

This manuscript describes development of a plasmonic nanoparticles platform for monitoring the effect of drugs on degradation of toxic amyloid beta ($A\beta$) oligomers. The conducted experiments and the outcomes almost validated the authors' claims about their approach. There are several comments that needs to be properly addressed by the authors to improve the robustness of this approach:

Our response: We appreciate your comments. We have thoroughly revised our manuscript based on your valuable feedback.

1- Any suitable assays that monitors the formation and degradation of $A\beta$ oligomers should be benign to the oligomerization and degradation processes. How the authors ensure that the degradation process is not affected by the plasmonic nanoparticles? What about the role of nanoparticles on the drugs and their degradation capacities? How the surface curvature of nanoparticles can affect the degradation process (as surface curvature can significantly affect biomolecule-nanoparticle interactions)?

Our response: We are grateful for your comment. To monitor the degradation of $A\beta$ oligomers on PNAC, we performed experiments such as high-resolution transmission electron microscopy (HRTEM) and gel electrophoresis.

Figure R1. HRTEM images of PNACs with $A\beta$ -degrading agents (a,b) EPPS (5 mM) or (c,d) protease XIV ($100 \mu\text{g ml}^{-1}$).

As shown in Fig. R1, the HRTEM images show the aggregation of PNACs treated by $A\beta$ -degrading agents such as EPPS

(5 mM) or protease XIV (100 $\mu\text{g mL}^{-1}$). In both the EPPS and the protease XIV cases, PNAC aggregation by A β degradation occurs. Specifically, slight aggregation of PNACs treated by EPPS occurs (Fig. R1a, b), while massive aggregation of PNACs treated by protease XIV occurs (Fig. R1c, d). This means that protein XIV degrades A β oligomers better than EPPS does, which is also supported by the UV-vis spectra (Fig. 3 and 4 in the manuscript). As such, the HRTEM analysis shows that PNACs are sensitive to differences in the degradation abilities among different A β -degrading agents.

We also performed gel electrophoresis as a suitable assay to ensure the degradation of the A β oligomer on the PNAC surface (Fig. R2). Before the gel running, bare AuNPs became aggregated because of their low stability in salt conditions (TAE 1X buffer condition); thus, no band is observed. In contrast, the PNACs samples exhibit a distinct band. This is attributed to the good dispersion of the PNACs (AuNPs with A β oligomers) because the A β oligomers provided them with good steric stability. With A β -degrading agents such as EPPS (5 mM and 50 mM) and protease XIV (50 $\mu\text{g/mL}$ and 100 $\mu\text{g/mL}$), the PNAC band is broadened, which implies that the aggregation of PNACs occurs because of the removal of A β oligomer from the PNACs. The aggregation of PNACs slows their motion in the agarose gel, resulting in band broadening.

Figure R2. Agarose gel electrophoresis of the PNAC sample depending on the treatment of A β -degrading agents in 1X TAE buffer solution. The difference in the color is due to the degradation of A β oligomer on PNAC surface (i.e., optical property changes because of AuNPs aggregation)

We agree with your view that the plasmonic nanoparticle itself can affect the A β degradation capability or process by the drug. For reliable drug screening, it is important to immobilize the A β oligomers on the AuNPs without protein denaturation. As you mentioned, the curvature of AuNPs is important in molecular interactions between AuNPs and proteins when an amyloid corona is formed^{1,2,3}. Woods et al.⁴ reported that the degree of curvature of AuNPs is important to prevent protein denaturation on the AuNP surface. In particular, they found that small-size AuNPs (~20 nm) have high degrees of curvature, and they interact with the amyloid protein without protein denaturation. Also, the 20-nm-sized AuNPs are suitable for monitoring the A β -degrading ability of various agents because the 20-nm-sized AuNPs can provide catalytic nucleation sites for A β ⁵, which can induce fast A β oligomerization on the AuNP surface. Moreover, 20-nm-sized AuNPs are widely used because of their stable optical properties and uniformity, which is important in increasing analytical sensitivity and reproducibility^{6,7}. The response to your comment regarding the curvature effect is

well described in our answer to your second question (Figure R3–R6). Please see Response 2.

We added this information to the revised manuscript (line 2 on page 3 and Supplementary Fig. S1 and S2).

2- To ensure the robustness of this approach, the authors are encouraged to reproduce their outcomes on various sizes of the plasmonic nanoparticles.

Our response: We are grateful for your comment. To investigate the possibility of drug screening on plasmonic nanoparticles of various sizes, we used 20-nm, 50-nm, and 100-nm AuNPs, as shown in Fig. R3.

Figure R3. **a**, SEM image of 20 nm AuNPs. **b**, Size distribution of 20 nm AuNPs (20.1 ± 1.24 nm, $N = 103$). **c**, SEM image of 50 nm AuNPs. **d**, Size distribution of 50 nm AuNPs (50.21 ± 4.05 nm, $N = 122$). **e**, SEM image of 100 nm AuNPs. **f**, Size distribution of 100 nm AuNPs (103.40 ± 7.43 nm, $N = 103$). The average size of the AuNPs was analyzed using ImageJ software.

We fabricated 20-nm, 50-nm, and 100-nm PNACs to reproduce our outcomes with A β -degrading agents (i.e., protease XIV). After the fabrication of various-sized PNACs, we performed a salt resistance test in a phosphate-buffered saline (PBS) solution (Fig. R4). As shown in Fig. R4a (the same as the Supplementary Fig. S7c), the self-aggregation of 20-nm

PNACs is negligible, indicating that the 20-nm PNACs are stable in salt conditions and thus suitable for use as colorimetric A β -degrading drug-screening platforms. By contrast, the absorbance peak of the 50-nm PNACs is slightly decreased because the 50-nm PNACs become somewhat aggregated in the PBS buffer. In the case of 100-nm PNACs, their absorbance peak is instantly decreased. The results imply that the PNACs become larger and that large-sized PNACs are unstable in salt condition.

Figure R4. **a**, The shift of the plasmonic peak of the 20 nm PNAC solution depending on the type of buffer solution (i.e., deionized water (DW) or phosphate buffer saline (PBS)). **b**, The shift of the plasmonic peak of the 50 nm PNAC solution in DW or PBS. **c**, The shift of the plasmonic peak of 100 nm PNAC solution in DW or PBS.

Moreover, we performed gel electrophoresis to analyze the uniformity of PNACs depending on the size of the core AuNPs (Fig. R5). The narrow and clear band of 20-nm PNACs indicates their high uniformity. By contrast, as the size of PNACs is increased, their band is broadened, indicating that the uniformity of large PNACs is poor. This is because large A β aggregates are easily generated as the size of the AuNPs increases. The results are consistent with the previous works^{8, 9, 10} reporting that large A β aggregates could be formed by increasing the size of AuNPs. Specifically, Kim et al⁸ investigated the degree of formation of A β aggregates with variously sized AuNPs using dark-field and TEM analysis. They found that 20-nm AuNPs had not aggregated and remained dispersed, while 50-nm and 80-nm AuNPs accelerated the formation of much larger A β aggregates.

Regarding single AuNPs, the hydrodynamic size of AuNPs is increased by A β aggregations on the AuNPs surface and larger AuNPs lead to more A β aggregation, thereby increasing local A β concentration around the AuNPs⁸. The increased local A β concentration at the surfaces of the AuNPs could enhance the probability of partially unfolded A β coming into frequent contact, resulting in more rapid clustering of AuNPs and A β ¹¹. This tendency is facilitated with large-sized AuNPs. The reaction between large-sized AuNPs and A β occurs rapidly and thereby interferes with the fabrication of uniform and stable PNACs. For these reasons, the uniformity of PNACs with large-sized core AuNPs is decreased, and their bands become broadened in gel electrophoresis (Fig. R5).

Figure R5. Agarose gel electrophoresis of the PNAC sample depending on the size of AuNPs in the 1X TAE buffer solution. The difference in the band color is due to the optical property changes depending on the size of AuNPs (i.e., 20, 50, and 100 nm). The experiment was performed twice for each condition.

With these variously sized nanoparticles, we conducted reproducibility experiments with protease XIV ($100 \mu\text{g mL}^{-1}$) in Fig. R6. When protease XIV is added to the 50-nm PNAC solution, the plasmonic peak of the 50-nm PNAC is shifted to longer wavelengths over time. This result indicates that $A\beta$ is removed from the PNAC surfaces, and consequently, the 50-nm PNAC solution is changed from red to purple (Fig. R6a, b). Also, the A_{609}/A_{525} is increased dramatically and becomes saturated within 5 min. Fig. R6c shows the decrease in the plasmonic peak wavelength of the 100-nm PNAC solution, which indicates that the 100-nm PNACs become aggregated. The A_{609} is decreased dramatically, and the reaction with protease XIV is saturated within 3 min. In contrast, as shown in Supplementary Fig. S6, the A_{609}/A_{525} of 20-nm PNAC is saturated over 40 min. These experimental results indicate that both the 50- and 100-nm PNACs are easily aggregated because of their instability, not just because of the removal of oligomers by protease XIV. Taken together, the use of 50-nm or 100-nm PNACs is not recommended for developing colorimetric $A\beta$ -degrading drug screening.

Figure R6. The protease XIV activity test with 50 and 100 nm PNAC-based platform. **a**, The UV-vis spectra of 50 nm

PNAC solution depending on the time with protease XIV ($100 \mu\text{g ml}^{-1}$). **b**, A_{609}/A_{525} shift of 50 nm PNAC solution with protease XIV ($100 \mu\text{g ml}^{-1}$). **c**, The UV-vis spectra of 100 nm PNAC solution depending on the time with protease XIV ($100 \mu\text{g ml}^{-1}$). **d**, A_{609} shift of 100 nm PNAC solution depending on the time with protease XIV ($100 \mu\text{g ml}^{-1}$). For your information, please be noted that we cannot use the parameter A_{609}/A_{525} because of no peak at around 525 nm in wavelength in the original spectra for 100 nm PNAC solution.

In conclusion, we chose the 20-nm AuNPs as a base material to fabricate reproducible PNACs for the following reasons: i) The 20-nm PNACs retain high steric stability, compared to that of the other PNACs; ii) the 20-nm AuNPs have good catalytic and optical properties⁵; iii) high uniformity is guaranteed in the synthesis of 20-nm PNACs.

All this information was added to our revised manuscript and supplementary information (lines 9–18 on page 3 & Supplementary Fig. S3, S4, and S5)

3- The multi layer corona formation may critically interfere with the degradation process. The authors may perform cryo-TEM and electron tomography to better interpret the formation of the amyloid corona at the surface of nanoparticles.

Our response: We are thankful for your comment. To investigate the formation of the amyloid corona, we cross-check the images of HRTEM and cryo-TEM (Fig. R7). In the cryo-TEM image, we observe that the AuNP is covered with a uniform-thickness amyloid corona layer. In detail, the cryo-TEM results show that the $A\beta$ layer of PNAC is ~ 3 nm in thickness. Both of the electron micrographs indicated that the thickness of the $A\beta$ layer of PNACs is similar to that of a single $A\beta$ oligomer reported previously¹². We are thus convinced that the amyloid corona of PNACs comprises a single-layered $A\beta$ oligomer. Using cryo-TEM (Tecnaï F20, FEI), we also tried electron tomography, but it failed. That is because the resolution becomes poor as the strong electron signal from the AuNPs impedes the signals of organic molecules like $A\beta$ oligomers.

Figure R7. The image of HRTEM (left, red), and cryo-TEM (right, blue)

Using cryo-TEM, we observe the degradation process of the amyloid corona on AuNPs over time (Fig. R8). In the absence of $A\beta$ -degrading agents (i.e., EPPS), the amyloid corona remains intact with approximately 3-nm thickness (Fig.

R8a). After the addition of EPPS (5 mM), the amyloid corona seems to decompose (Fig. R8b).

Figure R8. Cryo-TEM images of the process of the amyloid corona degradation process. **a-d**, Cryo-TEM images of PNAC without EPPS (**a**) or with 10 mM EPPS for 6 h (**b**), 12 h (**c**), and 24 h (**d**). **e**, The aggregation of PNACs after treatment of EPPS (10 mM) for 24 h.

After 6 h and 12 h, the amyloid corona on the AuNP surface is significantly decomposed (Fig. R8c); after 24 h, it almost disappears (Fig. R8d). Eventually, the degradation of the amyloid corona induces the massive aggregation of PNACs (Fig. R8e). In conclusion, our data support that the A β oligomers immobilized on the AuNPs are single-layered, and that they can be easily decomposed by A β -degrading agents (i.e., EPPS). This indicates that a PNAC-based colorimetric drug screening platform would be advantageous for monitoring the efficacy of A β -degrading drug candidates.

All this information was added to our revised manuscript and supplementary information (line 23–26 on page 3 & line 14–19 on page 4 & Supplementary Fig. S6)

Reviewer #2 (Remarks to the Author):

In this manuscript, authors describe a method called plasmonic nanoparticle amyloid corona (PNAC) to degrade the toxic amyloid β ($A\beta$) oligomers, which is a central feature of the onset and progression of Alzheimer's disease (AD), and monitor the drug-induced degradation of $A\beta$ oligomers by analyzing the colorimetric responses of PNACs. The authors used $A\beta$ -degrading proteases (protease XIV and MMP-9) and subsequently various small-molecule substances that have shown benefits in the treatment of AD. Although the drug screening technique has been shown to be meaningful in terms of specificity using plasmonic resonance peak shift, it is necessary to reinforce the interpretation from the scientific and clinical point of view, rather than a simple listing of various drug test results. To be published in this journal, clearer logical and authentic take-home messages are missing, and authors need to address following issues regardless of publishing another journal:

Our response: We appreciate your comments. We have carefully modified our manuscript based on your valuable feedback.

1. In p3, the authors wrote "First, we synthesized PNACs with AuNPs (20.1 ± 1.2 nm)." Why the size of AuNPs were chosen to 20 nm? It would be better if authors wrote interpretation of choosing this size of AuNPs. For example, this size was chosen because it is (1) the size of plasmonic peak resonance which can effectively differentiate absorbance peak of AuNPs and PNACs or (2) low aggregation condition or etc.

Our response: We are thankful to you for your comment. We chose 20-nm AuNPs as the base material to fabricate PNACs for several reasons. i) The 20-nm PNACs retain much better steric stability compared to that of other sized (50-nm and 100-nm) PNACs (Fig. R9 and Supplementary Fig. S7c); ii) High uniformity is guaranteed in synthesizing PNACs (Fig. R10); iii) 20-nm AuNPs have good catalytic and optical properties^{5,7}, which will help to precisely distinguish the differences in absorbance peaks between the PNACs under different conditions (type of drug, drug concentration, and treatment time).

To address the issue the reviewer noted, we fabricated 50- and 100-nm PNACs for comparison with 20-nm PNACs. After the fabrication of variously sized PNACs, we performed a salt resistance test in a PBS buffer (Fig. R9). The results show that the absorbance peak of 50-nm PNACs is slightly decreased because the 50-nm PNACs are somewhat aggregated in the PBS buffer, indicating that the $A\beta$ oligomer layer is not uniform on the 50-nm PNAC surface. In the case of 100-nm PNAC, the absorbance peak of 100 nm PNACs is instantly decreased. The results indicate that the PNACs have poorer steric stability and uniformity as they become larger. However, as shown in Fig. R9a, the aggregation of 20-nm PNACs is negligible, indicating that 20-nm PNACs are stable and suitable as colorimetric $A\beta$ -degrading drug screening platforms.

Figure R9. **a**, The shift of the plasmonic peak of the 20 nm PNAC solution depending on the type of buffer solution (i.e., deionized water (DW) or phosphate buffer saline (PBS)). **b**, The shift of the plasmonic peak of the 50 nm PNAC solution in DW or PBS. **c**, The shift of the plasmonic peak of 100 nm PNAC solution in DW or PBS.

Moreover, we performed gel electrophoresis to analyze the uniformity of PNACs depending on the size of core AuNPs (Fig. R10). In the case of 20-nm PNACs, a narrow and clear band is observed, indicative of the high uniformity of PNACs. As the size of the PNACs is increased, the band is broadened, revealing their poor uniformity. This result is consistent with the previous works that reported an increase in non-uniformity and instability of the amyloid corona owing to increases in nanoparticle size. Accordingly, we believe that the 20-nm PNACs are suitable for our strategy of colorimetric A β -degrading drug screening.

We added these reasons properly to the revised manuscript and supplementary information (lines 9–17 on page 3 & Supplementary Fig. S3, S4, and S5).

Figure R10. Agarose gel electrophoresis of the NAC sample depending on the size of AuNPs in the 1X TAE buffer solution. For your information, the difference in the band color is due to the optical property changes depending on the size of AuNPs (i.e., 20, 50, and 100 nm).

2. In this paper, ‘the relative absorbance ratio (A_{609} / A_{525})’ is used a lot. It is necessary to mention what the absorbance values of 609 nm and 525 nm mean.

Our response: We appreciate your comment. As shown in Fig. R11, the absorbance peak of the 20-nm AuNPs occurs at the wavelength of 520 nm. When A β is oligomerized on the AuNPs (PNAC), the absorbance peak of the PNACs is shifted to 525 nm. This is because surface plasmon resonance can be changed by the conjugation between plasmonic nanoparticles and proteins¹³. Therefore, we selected the absorbance peak at 525 nm (A_{525}) as a dispersive parameter. Similarly, we selected the absorbance peak at 609 nm (A_{609}) as a general aggregation parameter^{1,2}. For example, many studies have reported that when 20-nm AuNPs became aggregated in the aqueous state, the A_{609} increased. Thus, the use of the relative absorbance ratio (A_{609}/A_{525}) is very useful for quantifying the degree of AuNP aggregation.

All this information was added to our revised manuscript and supplementary information (line 22–26 on page 6).

Figure R11. Normalized absorbance spectrum of AuNP solution (black line) and PNAC solution (blue line). Inset: LSPR peak shift between the AuNP solution and PNAC solution.

3. Although the optimal Ab oligomer concentration was found in Fig. S2a, AuNP aggregation seems to be occurred about 50 % in Fig. 1d. However, only 5 nm peak shift occurred in Fig. S3. In the background of Fig. 1d, a significant amount of Ab fibrin particles estimated to be similar to the size of AuNPs were observed. Did this not affect the absorbance spectrum?

Our response: Thank you for your comment. We would like to clarify some misunderstandings. The PNACs seemed to aggregate in Fig. 1d because the PNAC solution was dried on a TEM grid, resulting in the unexpected binding of separated particles. This sometimes happens naturally in performing nanoparticle analysis with electron microscopy. Specifically, when the AuNP solution dried, various phenomena such as the coffee ring effect occurred¹⁴, whereby the nanoparticles could aggregate partly. It depends on the sampling conditions. For example, as shown in Fig. R12, most PNACs are well dispersed. By contrast, if the A β -degrading agents are added, a large three-dimensional aggregation of PNACs is observed, as shown in Fig. R13.

In Fig. 1d, the small bright particles in the background are not A β fibril particles but uranyl acetate. In general, uranyl acetate is used for negative staining to enhance the contrast between the background and samples. However, the main disadvantage of uranyl acetate for high-magnification studies is the granular microcrystalline nature of the dried stain, which appear similar to nanoparticles¹⁵. For this reason, uranyl acetate is observed to resemble small nanoparticles in HRTEM images (Fig. R12a). We also confirmed that these small nanoparticles were uranyl acetate because no small particles were observed in HRTEM images of materials with no staining (i.e., uranyl acetate) (Fig. R12b).

Figure R12. HRTEM image of PNACs with uranyl acetate staining (a), and without any staining (b).

Figure R13. HRTEM image of PNACs with A β -degrading agents (i.e., protease XIV (100 $\mu\text{g ml}^{-1}$))

4. In p5, the authors wrote “Physicochemical characterizations reveal that the PNACs are approximately 6 nm larger than the bare AuNPs (~22.45 nm) and possess slightly higher surface charges than bare AuNPs (Fig. 2c).”. However, since the absolute value of the zeta potential of PNAC is smaller than that of the Au bare NP, the expression of high surface charge is incorrect.

Our response: We appreciate your comment. To avoid an inaccurate description, we revised our manuscript as below:

Original sentence (page 6, line 3–4)

Physicochemical characterizations reveal that the PNACs are approximately 6 nm larger than the bare AuNPs (~22.45 nm) and possess slightly **higher** surface charges than bare AuNPs (Fig. 2c).

Revised sentence (page 6, line 3–4)

Physicochemical characterizations reveal that the PNACs are approximately 6 nm larger than the bare AuNPs (~22.45 nm) and possess slightly **reduced** surface charges relative to the bare AuNPs (Fig. 2c).

5. In this paper, only antibody tests were performed using graphene-based sensors, and no data on electrical conductance measurements related to drug screening. It is not necessary that the sensor was fabricated using graphene rather than general-purpose Si in order to find antibodies to immobilize the ab oligomer. If graphene is used, electrical conductance measurement data related to drug screening should be added to Fig. 6 in order to increase sensitivity than conventional sensors. It is recommended to send the contents of Fig 2 as supplementary as it may interfere with readers' understanding of the paper.

Our response: We appreciate your comment. The oligomeric nature of the A β conformation affects the significance and usability of PNACs. Our plan was to design A β oligomer-functionalized AuNPs as PNACs because A β oligomers are central features of the onset and progression of Alzheimer's disease (AD). Thus, it was very important for us to confirm the A β conformation type on the PNAC surface. In general, most researchers have used immunoreactivity tests to confirm the conformation of free A β particles or aggregates, as shown in Fig. R14. In our research, however, it is impossible to use this conventional method for verifying A β conformation type on the PNAC because of severe fluorescence quenching in the presence of the AuNPs^{16, 17, 18}.

Figure R14. **a**, Kinetic analysis of oligomer specific immunoreactivity during fibrillogenesis. **b**, Immunoreactivity test of each A β conformation (i.e., A β monomer, A β oligomer and A β fibril)^{19, 20}

Alternatively, we tried to use reduced graphene oxide (rGO)-based sensors that we have developed and researched for years. rGO is a promising nanomaterial possessing high conductivity, numerous reaction sites, and high reproducibility^{21, 22}. We have investigated graphene-based point-of-care platforms and verified versatile applications of rGO. We reported i) a wafer-scaled, reproducible rGO-based biosensor for the detection of A β ²³, ii) a neural-derived exosomal A β detection platform through enhancement of functionalization efficiency using O₂-plasma²⁴, and iii) enzyme-modified graphene-based monitoring of drug effects²⁵. Based on these techniques, we developed rGO-based sensors for revealing the conformation of A β corona on PNAC. Specifically, we immobilized conformational-specific antibodies (6E10, A11, and OC antibody) onto the graphene-based sensor using 1-pyrenebutanoic acid, succinimidyl ester (PBSE). When PNAC was attached to a specific conformational antibody, the relative resistance was changed because AuNPs are electron transporters^{26, 27}. In conclusion, we found that the conformation of A β on the PNAC was A β oligomer using these graphene-based sensor (Fig. 2a, b). Please note that the use of graphene-based sensors in this manuscript is not for graphene-based drug screening, but for revealing the conformation of A β corona. Our drug-screening method in the manuscript is a colorimetric method using the plasmonic phenomena of AuNPs, as stated in the manuscript title. If the content of graphene-based drug screening is added, we are concerned that the unity of our manuscript could be damaged. Very recently, for another research project in our laboratory, we have begun development of a drug-screening platform using a graphene-based field-effect transistor platform (gFET) (Fig. R15).

Figure R15. Schematic diagram of the graphene-based drug screening platform and working principle. **a**, The identification of PNAC drug degradation by immobilization of bio-receptor (i.e., antibodies or aptamers). **b**, Real-time monitoring of A β degradation by direct immobilization of PNAC on the graphene surface.

To develop a gFET-based drug screening platform, we devised two strategies (Fig. R15).

- 1) Identification of PNAC drug degradation by immobilization of bio-receptors (i.e., antibodies or aptamers)
- 2) Real-time monitoring of A β degradation by direct immobilization of PNAC on the graphene surface

Of the two, we chose the second strategy because it is simpler and more efficient as a real-time drug-screening platform than the first method. In detail, the method of molecular-based immobilization has a simpler protocol than the method of antibody immobilization, so the variables are reduced when experimenting. From this perspective, we proceeded with preliminary testing using a molecular-based immobilization method to monitor the real-time efficacy of anti-A β drug candidates (EPPS). In detail, as the amyloid corona decomposes, the surface of the electron-rich AuNPs is revealed, allowing the transfer of electrons from the AuNPs. This phenomenon causes the n-doping effect of the gFET and a leftward shift in the V_g - I_{ds} transfer curve. Based on this principle, previous studies have reported gFET using the G-quadruplex conformation of aptamer^{28,29}, gFET for miRNA detection using PNA³⁰, and extracellular vesicle detection³¹.

Fig. R16a shows the SEM image of well-immobilized PNACs on the rGO surface where PBSE is functionalized. In contrast, bare AuNPs rarely interact with the PBSE-functionalized rGO surface (Fig. R16b) because AuNPs cannot bind to PBSE.

Figure R16. SEM image of rGO surfaces after incubation with the PNAC (a) and bare AuNP (b).

Next, we evaluated the feasibility of the PNAC-immobilized gFET as a drug-screening platform by treating with EPPS. We measured the transfer curves of gFET every 10 min to assess A β degradation caused by the EPPS treatment. The Ag/AgCl reference electrode was used as a liquid-gate electrode, and the gate voltage was swept from -0.5 V to 1 V. Fig. R17 shows the V_g - I_{ds} transfer curves of the PNAC-immobilized gFET with or without EPPS.

Figure R17. Transfer characteristic curves measured when the PNAC-immobilized gFET was exposed to without (a) and with EPPS (b) in PBS buffer. The transfer curves were measured every 10 minutes after each solution was injected into the micro-channel embedded in our customized jig system. c, The V_{Dirac} variation of PNAC-immobilized gFET w/ or w/o EPPS.

The transfer curves show the intrinsic ambipolar characteristics around the Dirac point (V_{Dirac}), which indicates the charge neutrality point. When a PBS buffer without EPPS is added, no significant shift of V_{Dirac} and transfer characteristic curves of gFET occurs. Meanwhile, the n-doping effect occurs when the EPPS solution (5 mM) is introduced to the PNAC-immobilized gFET. These results imply that the amyloid corona of PNACs is degraded by EPPS, and the PNAC-immobilized gFET is a promising drug-screening platform for monitoring the efficacy of A β -degrading agents. This is just a preliminary work to test the feasibility of the PNAC-immobilized gFET as a drug-screening platform. It will take at least six months to complete the full set of experiments. We are planning to publish them in any journal in the future. Thus, we

would like to provide only the reviewer with this preliminary data regarding PNAC-immobilized gFET.

As mentioned previously, our drug-screening method in the manuscript title is a colorimetric method using the plasmonic phenomena of AuNPs. The use of a graphene-based sensor was only intended to verify the A β conformation of PNACs, not for drug screening. If content regarding graphene-based drug screening is added, we are deeply concerned that the unity of our manuscript could be damaged. We ask that the reviewer understands this situation.

Furthermore, we considered your recommendation for moving the contents of Fig. 2 to the supplementary information. We suggest keeping Fig. 2b-e in the main manuscript because these data are essential for understanding the nature of PNACs. However, Fig. 2a may interfere with readers' understanding of this manuscript, because it depicts a photograph of a graphene sensor. Accordingly, the previous Fig. 2a was sent to Supplementary Fig. S10. Instead, we recomposed Fig. 2a for a better understanding of our method used to verify the conformation of the A β corona on PNAC. The logic suggested in the new Fig. 2a will help to decipher the response of the antibody-immobilized graphene-based sensor, as shown in Fig. 2b.

To clarify this in our manuscript, we have added detailed information in the revised manuscript (line 21–24 on page 4 and Fig. 2a) and Supplementary Fig. S10–S11.

6. Why did dose- and time-dependent tests run on small molecules (EPPS), not on proteins (Protease XIV, MMP-9)?
Protease?

Our response: We appreciate your comment. Following your feedback, all the data with enzymes (protease XIV and MMP-9) in our manuscript are reanalyzed by a dose-dependent test. In specific, we analyze the relative absorbance (T_1-T_0) as a function of the protease XIV concentration using the sigmoidal dose-response model (Fig. R18a).

$$\text{The relative absorbance } (T_1-T_0) = \frac{1}{1 + e^{-x}}, R^2 = 0.97$$

The values of EC₅₀ and maximal efficacy are estimated as 29.83 $\mu\text{g mL}^{-1}$ and 1.023 (a.u.), respectively.

In the case of MMP-9, we extract the values of the half-maximal effective concentration EC₅₀ and the maximal efficacy by the sigmoidal dose-response model, following the equation (Fig. R18b)

$$\text{The relative absorbance } (T_1-T_0) = \frac{1}{1 + e^{-x}} + 0.03607, R^2 = 0.93$$

The values of EC₅₀ and maximal efficacy are estimated as 1.283 pg mL^{-1} and 0.4213 (a.u.), respectively.

Figure R18. Dose-response analysis of A β -degrading enzyme (protease XIV and MMP-9). **a**, The relative absorbance (T_1-T_0) of PNAC solution with various concentrations of protease XIV. **b**, The relative absorbance (T_1-T_0) of PNAC solution with various concentrations of MMP-9.

We acknowledge that time-dependent tests of such enzymes are important; we have added a protease XIV kinetics test, as shown in Fig. R18. In detail, we added 25, 50, 100, and 200 $\mu\text{g mL}^{-1}$ of protease XIV to the PNAC solutions to investigate its time-variable effects. Then, we performed a kinetic analysis of protease XIV depending on degradation times reaching 1 h with the four different concentrations of protease XIV. As the concentration of protease XIV increased, the time to reach 50% of A β -degrading drug activity ($\text{Time}_{50\%}$) and the time constant (τ) exponentially decreased (Fig. R19), i.e.:

$$\text{Time}_{50\%} = 22.13 \times \left[\frac{1}{\text{Protease XIV concentration}} \right] + 3.939 \quad (R^2 = 0.96)$$

$$\text{Time constant } (\tau) = 31.94 \times \left[\frac{1}{\text{Protease XIV concentration}} \right] + 5.684 \quad (R^2 = 0.96)$$

Moreover, we investigated the time-variable effects of MMP-9 by performing a kinetic analysis of MMP-9 depending on degradation time up to 1 h with four different MMP-9 concentrations. As the MMP-9 concentration was increased, the time to reach 50% of A β -degrading drug activity ($\text{Time}_{50\%}$) and time constant (τ) were exponentially decreased (Supplementary Fig. 15d), i.e.:

$$\text{Time}_{50\%} = 54.24 \times \left[\frac{1}{\text{MMP-9 concentration}} \right] + 0.7842 \quad (R^2 = 0.99)$$

$$\text{Time constant } (\tau) = 78.46 \times \left[\frac{1}{\text{MMP-9 concentration}} \right] + 1.228 \quad (R^2 = 0.98)$$

The results show that dose- and time-dependent proteolytic activity of A β -degrading protease (i.e., MMP-9) can be measured by our screening platform with high sensitivity (i.e., 100 fg mL^{-1} analytical sensitivity).

We added this information in the revised manuscript (line 8–18 on page 7 & line 2–13 on page 8) and in Supplementary Fig. S15.

Figure R19. Kinetic analysis of A β -degrading enzyme (i.e., protease XIV and MMP-9). **a**, A_{609}/A_{525} of PNAC solution was measured for 1 h with various concentrations of protease XIV. **b**, The kinetic analysis of A β degradation depending on the concentration of protease XIV. **c**, A_{609}/A_{525} of PNAC solution was measured for 1 h with various concentrations of MMP-9. **d**, The kinetic analysis of A β degradation depending on the concentration of MMP-9.

7. In Fig. 4d, the authors introduced Half-life and Time constant values for prediction of appropriated drug treatment time as mentioned in p8 “These results help to predict the appropriate treatment time depending on the EPPS concentration.”. However, the specific clinical usage of these values was not written, and it is difficult to understand the difference between them.

Our response: We appreciate your comment. Theoretically, there are three mechanisms of anti-A β drugs: i) solubilization by direct binding to A β , ii) phagocytosis by activated microglia, and iii) activation of A β extraction to the blood by the peripheral sink. Among these mechanisms, it is known that EPPS follows the solubilization mechanism by direct binding to A β ³². Therefore, we applied the binding kinetics model to studying the drug behavior of EPPS³³. Using the binding kinetics model, we quantified the drug-receptor (A β) binding kinetics by the extraction of a time constant (τ). The rate constant is the inverse of the time constant ($1/\tau$), which is important in investigating the relationship between the drug structure and the drug-receptor binding kinetics³⁴. Detailed information regarding the time constant should provide

a handle for the rational drug optimization of the kinetics binding profile of drug candidates³⁵.

We consider that the term of half-life can induce misunderstandings because it is usually used to refer to the dissociation half-life of drugs in the body after drug uptake. Therefore, we replaced the term ‘half-life’ with ‘Time_{50%}’ which represents the time to reach 50% of drug activity³⁶.

The half-life (Time_{50%}) is associated with the optimal dose selection. In general, the optimal drug dose selection through the establishment of exposure–response relationships can improve the safety and efficacy of drugs. These exposure–response relationships can be initially evaluated using *in vitro* and *in vivo* systems. This helps in defining the optimal doses that can be tested in early clinical trials³⁷. Without data generated from the early stage of drug development, clinical trials can sometimes lead to the selection of suboptimal dosage regimens, which can cause their failure³⁸.

Figure R20. Time-dependent drug activity of Jasplakinolide. **a**, Activity of Jasplakinolide depending on the various target concentration. **b**, The time to reach 50% of the drug activity³⁶.

To avoid this situation, many researchers have developed sophisticated approaches to integrate exposure–time (pharmacokinetic) with exposure–effect (pharmacodynamics) information³⁸. For example, researchers have investigated *in vitro* time-dependent drug activity for various drugs such as Jasplakinolide³⁶, Doxorubicin³⁹, and many other drugs⁴⁰. As shown in Fig. R20, time-dependent drug activity is acquired to investigate the time to reach 50% of drug activity (Time_{50%}). Our data for Time_{50%} may help to select the optimal concentrations of drug candidates for successful early clinical trials³⁸.

To ensure clarity in our manuscript, we have added this information (line 18–20 on page 9).

8. In Fig. 3a and 3h using high doses of protein (Protease XIV, MMP-9), the absorbance peaks changed significantly, indicating that the AuNP array was broken. However, in Fig. 4a using small molecules (EPPS), only small peak shifts due to degradation are observed. The authors should provide a concrete interpretation from a scientific point of view and Alzheimer's disease (AD) treatment point of view.

Our response: Thank you for your comment. As EPPS follows the solubilization mechanism by directly binding to A β , as mentioned above in our answer to your 7th comment³², EPPS dismantles the β -stacking inside A β oligomers and cuts

them into individual monomers^{32, 41}. Meanwhile, protease XIV and MMP-9 continuously degrade A β into numerous small peptide fragments by proteolysis until the termination of the enzyme-substrate reaction^{42, 43}. For example, protease XIV can cleave many regions of the peptide chain of the monomeric A β sequence, as shown in Fig. R21, and thereby breaks A β oligomers down into numerous small peptide fragments. Both protease XIV and MMP-9 perfectly eliminate A β oligomers on the surface. Therefore, A β degradation with the proteases can occur rapidly on the PNAC surface, resulting in a dramatic peak shift of the PNAC solution, compared to that of EPPS.

Figure R21. Model of enzymatic digestion and crystal structures of A β . **a**, Enzymatic digestion patterns of protease XIV. **b**, the structure of 5-mer of A β peptides with cross- β structure. **c**, Enlargement of cross- β structure around K28 in panel **b**⁴².

To clarify this in our manuscript, we have added detailed information in the revised manuscript (line 21–25 on page 8 & line 1–5 on page 9)

Minor issues:

1. In Fig. 1b, 1c (inset), 1d (inset), scale bars are missing.

Our response: Thank you for your comment. We have added scale bars to Fig. 1b-1d.

2. In Fig. S2b and S2c, the authors should write the legend of DI and PBS.

Our response: We appreciate your comment. We added the legend of DI and PBS, which are deionized water and phosphate-buffered saline, respectively.

3. In Fig. 3b, the authors should clearly indicate on the graph or caption whether cases (1)-(5) were overlapped.

Our response: Thank you for your comment. To ensure the clarity of our manuscript, we noted that the spectra are overlapped in the figure caption in the manuscript.

4. In Fig. 3a, 3f-3l, 4a, 5a-5e, throughout the graph, the authors did not use common line colors, and the direction of the arrows, not marked with a clear legend, can cause confusion for the readers.

Our response: We are grateful for your comment. We acknowledge that the arrow could cause confusion for readers. We have eliminated all the arrows in Fig. 3a, 3f-3l, 4a, and 5a-5e. The line colors of Fig. 3–5 were changed from old colors to MATLAB’s current colors (Fig. R22).

Current color		Old color	
	[0, 0.4470, 0.7410]		[0, 0, 1]
	[0.8500, 0.3250, 0.0980]		[0, 0.5, 0]
	[0.9290, 0.6940, 0.1250]		[1, 0, 0]
	[0.4940, 0.1840, 0.5560]		[0, 0.75, 0.75]
	[0.4660, 0.6740, 0.1880]		[0.75, 0, 0.75]
	[0.3010, 0.7450, 0.9330]		[0.75, 0.75, 0]
	[0.6350, 0.0780, 0.1840]		[0.25, 0.25, 0.25]

Figure R22. Default colors in 2D graphs used in MATLAB (MathWorks, MA, USA).

References

1. Wu S-H, Wu Y-S, Chen C-h. Colorimetric sensitivity of gold nanoparticles: minimizing interparticular repulsion as a general approach. *Analytical chemistry* 2008, **80**(17): 6560-6566.
2. Zhou Y, Dong H, Liu L, Xu M. Simple Colorimetric Detection of Amyloid β peptide (1–40) based on Aggregation of Gold Nanoparticles in the Presence of Copper Ions. *Small* 2015, **11**(18): 2144-2149.
3. Gao G, Zhang M, Gong D, Chen R, Hu X, Sun T. The size-effect of gold nanoparticles and nanoclusters in the inhibition of amyloid- β fibrillation. *Nanoscale* 2017, **9**(12): 4107-4113.
4. Woods KE, Perera YR, Davidson MB, Wilks CA, Yadav DK, Fitzkee NC. Understanding protein structure deformation on the surface of gold nanoparticles of varying size. *The Journal of Physical Chemistry C* 2016, **120**(49): 27944-27953.
5. Choi I, Lee LP. Rapid detection of A β aggregation and inhibition by dual functions of gold nanoplasmonic particles: catalytic activator and optical reporter. *ACS nano* 2013, **7**(7): 6268-6277.
6. Huang X, El-Sayed MA. Gold nanoparticles: Optical properties and implementations in cancer diagnosis and photothermal therapy. *Journal of advanced research* 2010, **1**(1): 13-28.
7. Jans H, Huo Q. Gold nanoparticle-enabled biological and chemical detection and analysis. *Chemical Society Reviews* 2012, **41**(7): 2849-2866.
8. Kim Y, Park J-H, Lee H, Nam J-M. How do the size, charge and shape of nanoparticles affect amyloid β aggregation on brain lipid bilayer? *Scientific reports* 2016, **6**: 19548.
9. Liu J, Legros S, Ma G, Veinot JG, Von der Kammer F, Hofmann T. Influence of surface functionalization and particle size on the aggregation kinetics of engineered nanoparticles. *Chemosphere* 2012, **87**(8): 918-924.
10. John T, Gladysz A, Kubeil C, Martin LL, Risselada HJ, Abel B. Impact of nanoparticles on amyloid peptide and protein aggregation: a review with a focus on gold nanoparticles. *Nanoscale* 2018, **10**(45): 20894-20913.
11. Mahmoudi M, Kalhor HR, Laurent S, Lynch I. Protein fibrillation and nanoparticle interactions: opportunities and challenges. *Nanoscale* 2013, **5**(7): 2570-2588.
12. Ahmed M, Davis J, Aucoin D, Sato T, Ahuja S, Aimoto S, *et al.* Structural conversion of neurotoxic amyloid- β 1–42 oligomers to fibrils. *Nature structural & molecular biology* 2010, **17**(5): 561.
13. Park H-J, Vak D, Noh Y-Y, Lim B, Kim D-Y. Surface plasmon enhanced photoluminescence of conjugated polymers. *Applied physics letters* 2007, **90**(16): 161107.
14. Zhang Z, Zhang X, Xin Z, Deng M, Wen Y, Song Y. Controlled inkjetting of a conductive pattern of silver nanoparticles based on the coffee ring effect. *Advanced Materials* 2013, **25**(46): 6714-6718.
15. De Carlo S, Harris JR. Negative staining and cryo-negative staining of macromolecules and viruses for TEM. *Micron* 2011, **42**(2): 117-131.
16. Dulkeith E, Morteani A, Niedereichholz T, Klar T, Feldmann J, Levi S, *et al.* Fluorescence quenching of dye molecules near gold nanoparticles: radiative and nonradiative effects. *Physical review letters* 2002, **89**(20): 203002.
17. Xue C, Xue Y, Dai L, Urbas A, Li Q. Size and shape dependent fluorescence quenching of gold nanoparticles on perylene dye. *Advanced Optical Materials* 2013, **1**(8): 581-587.
18. Lee S, Cha EJ, Park K, Lee SY, Hong JK, Sun IC, *et al.* A near infrared fluorescence quenched gold nanoparticle imaging probe for in vivo drug screening and protease activity determination. *Angewandte Chemie International Edition* 2008, **47**(15): 2804-2807.

19. Kaye R, Head E, Sarsoza F, Saing T, Cotman CW, Necula M, *et al.* Fibril specific, conformation dependent antibodies recognize a generic epitope common to amyloid fibrils and fibrillar oligomers that is absent in prefibrillar oligomers. *Molecular neurodegeneration* 2007, **2**(1): 18.
20. Kaye R, Head E, Thompson JL, McIntire TM, Milton SC, Cotman CW, *et al.* Common structure of soluble amyloid oligomers implies common mechanism of pathogenesis. *Science* 2003, **300**(5618): 486-489.
21. Bianco A, Cheng H-M, Enoki T, Gogotsi Y, Hurt RH, Koratkar N, *et al.* All in the graphene family—A recommended nomenclature for two-dimensional carbon materials. Elsevier; 2013.
22. Ivanovskii AL. Graphene-based and graphene-like materials. *Russian Chemical Reviews* 2012, **81**(7): 571.
23. Kim J, Chae M-S, Lee SM, Jeong D, Lee BC, Lee JH, *et al.* Wafer-scale high-resolution patterning of reduced graphene oxide films for detection of low concentration biomarkers in plasma. *Scientific reports* 2016, **6**(1): 1-8.
24. Chae M-S, Kim J, Jeong D, Kim Y, Roh JH, Lee SM, *et al.* Enhancing surface functionality of reduced graphene oxide biosensors by oxygen plasma treatment for Alzheimer's disease diagnosis. *Biosensors and Bioelectronics* 2017, **92**: 610-617.
25. Chae M-S, Yoo YK, Kim J, Kim TG, Hwang KS. Graphene-based enzyme-modified field-effect transistor biosensor for monitoring drug effects in Alzheimer's disease treatment. *Sensors and Actuators B: Chemical* 2018, **272**: 448-458.
26. Thelander C, Magnusson MH, Deppert K, Samuelson L, Poulsen PR, Nygård J, *et al.* Gold nanoparticle single-electron transistor with carbon nanotube leads. *Applied Physics Letters* 2001, **79**(13): 2106-2108.
27. Mao S, Lu G, Yu K, Bo Z, Chen J. Specific protein detection using thermally reduced graphene oxide sheet decorated with gold nanoparticle antibody conjugates. *Advanced materials* 2010, **22**(32): 3521-3526.
28. Wang Z, Hao Z, Yu S, De Moraes CG, Suh LH, Zhao X, *et al.* An Ultraflexible and Stretchable Aptameric Graphene Nanosensor for Biomarker Detection and Monitoring. *Advanced Functional Materials* 2019, **29**(44): 1905202.
29. Hao Z, Wang Z, Li Y, Zhu Y, Wang X, De Moraes CG, *et al.* Measurement of cytokine biomarkers using an aptamer-based affinity graphene nanosensor on a flexible substrate toward wearable applications. *Nanoscale* 2018, **10**(46): 21681-21688.
30. Cai B, Huang L, Zhang H, Sun Z, Zhang Z, Zhang G-J. Gold nanoparticles-decorated graphene field-effect transistor biosensor for femtomolar MicroRNA detection. *Biosensors and Bioelectronics* 2015, **74**: 329-334.
31. Wu D, Zhang H, Jin D, Yu Y, Pang D-W, Xiao M-M, *et al.* Microvesicle detection by a reduced graphene oxide field-effect transistor biosensor based on a membrane biotinylation strategy. *Analyst* 2019, **144**(20): 6055-6063.
32. Kim HY, Kim HV, Jo S, Lee CJ, Choi SY, Kim DJ, *et al.* EPPS rescues hippocampus-dependent cognitive deficits in APP/PS1 mice by disaggregation of amyloid- β oligomers and plaques. *Nature communications* 2015, **6**(1): 1-14.
33. Tonge PJ. Drug–target kinetics in drug discovery. *ACS chemical neuroscience* 2018, **9**(1): 29-39.
34. Pan AC, Borhani DW, Dror RO, Shaw DE. Molecular determinants of drug–receptor binding kinetics. *Drug discovery today* 2013, **18**(13-14): 667-673.
35. Huggins DJ, Sherman W, Tidor B. Rational approaches to improving selectivity in drug design. *Journal of medicinal chemistry* 2012, **55**(4): 1424-1444.
36. Bubb MR, Spector I, Beyer BB, Fosen KM. Effects of jasplakinolide on the kinetics of actin polymerization an explanation for certain in vivo observations. *Journal of Biological Chemistry* 2000, **275**(7): 5163-5170.

37. Overgaard R, Ingwersen S, Tornøe C. Establishing good practices for exposure–response analysis of clinical endpoints in drug development. *CPT: pharmacometrics & systems pharmacology* 2015, **4**(10): 565-575.
38. Butterfield J, Lodise Jr TP, Pai MP. Applications of Pharmacokinetic and Pharmacodynamic Principles to Optimize Drug Dosage Selection. *Therapeutic Drug Monitoring: Newer Drugs and Biomarkers* 2012: 175.
39. Farhane Z, Bonnier F, Howe O, Casey A, Byrne HJ. Doxorubicin kinetics and effects on lung cancer cell lines using in vitro Raman micro spectroscopy: binding signatures, drug resistance and DNA repair. *Journal of biophotonics* 2018, **11**(1): e201700060.
40. Filppula AM, Parvizi R, Mateus A, Baranczewski P, Artursson P. Improved predictions of time-dependent drug-drug interactions by determination of cytosolic drug concentrations. *Scientific reports* 2019, **9**(1): 1-14.
41. Kim HY, Kim Y, Han G, Kim DJ. Regulation of in vitro A β 1-40 aggregation mediated by small molecules. *Journal of Alzheimer's Disease* 2010, **22**(1): 73-85.
42. Numata K, Kaplan DL. Mechanisms of enzymatic degradation of amyloid β microfibrils generating nanofilaments and nanospheres related to cytotoxicity. *Biochemistry* 2010, **49**(15): 3254-3260.
43. Saido T, Leissring MA. Proteolytic degradation of amyloid β -protein. *Cold Spring Harbor perspectives in medicine* 2012, **2**(6): a006379.

Reviewer #1 (Remarks to the Author):

The authors have thoroughly considered my comments by conducting a fair amount of new experiments. I would publish this interesting paper.

Reviewer #3 (Remarks to the Author):

This study presents a potentially effective assay to detect the disaggregation activity of anti-amyloid compounds.

It would seem misleading to call oligomers the aggregates in the corona. The manuscript does not present convincing evidence that these aggregates are oligomers. Oligomers have a few nm size in all dimensions, not just along the thickness, as shown by cryo-TEM in this work.

The antibodies used (6E10, A11 and OC) also do not provide conclusive evidence that the assemblies in the corona are oligomers because the results of these assays are consistent also with the presence of larger (in the two-dimensions of the surface) assemblies. I would encourage the authors to check this by carrying out a control experiments to reproduce the table in Fig. 2a with purified monomers, oligomers and fibrils.

Importantly, the authors should provide evidence that the assemblies in the corona resemble the oligomers in human brains, or at least the oligomers produced in vitro where the aggregation process is homogeneous and spontaneous.

It would be interesting to extend the PNAC assay to detect not just the disaggregation activity, but also the anti-aggregation activity of candidate compounds. This could be achieved by adding the compounds during the formation of the corona.

Overall, there are many assays to measure anti-aggregation activity of candidate compounds. The novelty would be to introduce a new assay for oligomer-specific compounds. Not enough evidence is presented that the present study achieves this goal.

Response Letter

Reviewer #3 (Remarks to the Author):

This study presents a potentially effective assay to detect the disaggregation activity of anti-amyloid compounds.

It would seem misleading to call oligomers the aggregates in the corona. The manuscript does not present convincing evidence that these aggregates are oligomers. Oligomers have a few nm size in all dimensions, not just along the thickness, as shown by cryo-TEM in this work.

Our response: We appreciate your comments. We are certain that the amyloid corona on the PNAC consist of individual A β oligomers. To address this issue, we have experimented thoroughly and revised our manuscript based on your valuable feedback.

The antibodies used (6E10, A11 and OC) also do not provide conclusive evidence that the assemblies in the corona are oligomers because the results of these assays are consistent also with the presence of larger (in the two-dimensions of the surface) assemblies. I would encourage the authors to check this by carrying out a control experiments to reproduce the table in Fig. 2a with purified monomers, oligomers and fibrils.

Our response: We are grateful for your comment. To perform the control experiments, as you suggested, we synthesized A β oligomers and fibrils from a purified A β monomer prepared by treatment with hexafluoro-2-propanol, which is known as a β -sheet breaker¹. As shown in Fig. R1, we confirmed that the purified A β monomer is well dispersed with a uniform size of \sim 320 pm.

Figure R1. AFM characterization of A β monomers. **a**, A representative AFM image over a $5 \times 5 \mu\text{m}^2$ area of the purified A β monomers. **b**, A magnified image from **a**. **c**, The cross-sectional profile of the single A β monomer indicated in **b**.

To prepare the A β oligomers, we incubated the A β monomer solution in phosphate-buffered saline

(PBS, pH 7.4) for 24 h at 4 °C. Then we used the sucrose-gradient centrifugation method² to extract the purified A β oligomer from the incubated A β solution (Fig. R2a–d). In detail, the A β samples were mixed together and 3-mL aliquots were injected into a sucrose step gradient (10–40% with an increment of 10%). The sucrose solutions were centrifuged for 12 h at 23,000g in an SW 70 Ti swinging-bucket rotor (Beckmann) at room temperature (22 °C).

The results showed a significant increase in the average height of the A β species as a function of the sucrose fraction density (Fig. R2e); a drastic increase in the cross-sectional diameter (i.e., height) of the A β species was also found in the 40% fraction. The average sizes of the A β species from the 10, 20, 30, and 40% sucrose fractions are 1.64, 2.50, 2.23, and 7.28 nm, respectively.

The following data supports our conclusion that A β oligomers are formed in the 20% sucrose fraction:

- i) The 30% sucrose fraction (Fig. R2c) shows A β protofibrils as well as oligomers.
- ii) The average size of A β species in the 20% sucrose fraction is quite similar to that found in the amyloid corona on the PNACs.

Figure R2. AFM characterization of A β aggregates present in sucrose fractions. a–d, Representative AFM images $5 \times 5 \mu\text{m}^2$ of the A β aggregates present in different fractions of the sucrose gradient. **e,** Average heights of A β aggregates present in the different sucrose fractions. As the concentration of sucrose increases, the mean height of individual aggregates tends to increase.

To prepare purified A β fibrils, we incubated the purified A β solution in a 10-mM hydrochloric acid

(HCl) solution with the final A β solution concentration of 100 μ M. The solution was incubated in a thermomixer (Eppendorf, Germany) at 37 $^{\circ}$ C for 3 days. We then dialyzed the solution against distilled water for 48 h by using 50-kDa dialysis membranes (Biovision Inc, USA) to remove extra A β monomers and small aggregates. The final products were analyzed by AFM (Fig. R3).

Figure R3. AFM images of purified A β fibrils (image size: 5 \times 5 μ m²).

Figure R4. Performance test of the graphene-based sensor with three different conformation-specific antibodies. **a–c**, Representative AFM images of A β monomers, oligomers, and fibrils, respectively. **d**, Schematic of performance test of the graphene-based sensor with three different conformation-specific antibodies (i.e., 6E10, A11, and OC). **e**, A heatmap of relative resistance changes of the antibody-immobilized graphene sensor depending on the antibody treatment of each purified A β solution.

As a control experiment to reproduce the table in Fig. 2a, as suggested by the reviewer, we utilized graphene-based sensors wherein each surface of the sensor was functionalized with three different types of antibodies (monoclonal 6E10, polyclonal A11, and polyclonal OC). We treated the graphene sensors with each of the purified A β solution (i.e., purified A β monomers, oligomers, and fibrils) (Fig. R4a–c) and monitored the relative resistance changes ($\Delta R/R$, %) of each sensor with the three different types of antibodies (Fig. R4d). The results indicated that each graphene sensor interacted with specificity for each purified A β species. The results are expressed as a heatmap (Fig. R4e), where green represents greater interaction between each antibody and the corresponding conformation of A β while red means no interaction between them. The results are exactly consistent with the table in Fig. R4d and Fig. 2a in the manuscript. In conclusion, our additional experimentation confirmed that the amyloid corona on PNAC comprises single A β oligomers.

We added this information to the revised manuscript (lines 15–20 on page 5 and Supplementary Fig. 13–16).

Importantly, the authors should provide evidence that the assemblies in the corona resemble the oligomers in human brains, or at least the oligomers produced in vitro where the aggregation process is homogeneous and spontaneous.

Our response: We appreciate your comments. To confirm whether the corona of the PNACs resembles the oligomers present in human brains, we synthesized A β oligomers in human cerebral spinal fluid (hCSF, Lee Biotech, USA). In detail, we first incubated the A β monomers in hCSF for 24 h at 4 °C to fabricate A β oligomers^{1,3}. Then, we mixed the hCSF-A β oligomer with a concentrated AuNP solution. The mixed solution was incubated for 24 h in a thermomixer (Eppendorf, Germany) that controlled the incubation conditions to 37 °C with 120 Hz shaking. This yielded PNACs in hCSF (i.e., hCSF-PNACs). To confirm whether the hCSF-PNACs has characteristics similar to those of PNACs made in PBS (i.e., PBS-PNACs), we performed high-resolution transmission electron microscopy (HRTEM) analysis, graphene-based conformation testing, and electrophoresis with hCSF-PNACs (Fig. R5). Moreover, we verified the drug screening performance with hCSF-PNACs using protease XIV (P14) and A β -degrading 4-(2-hydroxyethyl)-1-piperazinepropanesulphonic acid (EPPS), as shown in Fig. R6 and R7.

Figure R5. hCSF-PNAC characterization. **a**, HRTEM image of hCSF-PNAC. **b**, The relative resistance changes of the antibody-immobilized graphene sensor after treatment with bare AuNPs (orange), PBS-PNACs (wine color), and hCSF-PNACs (green). The bars in (b) represent the average \pm standard deviation calculated from five independent graphene-based sensors. **c**, Agarose gel electrophoresis of the PBS-PNAC and hCSF-PNAC samples in the $1\times$ Tris-acetate/ethylenediaminetetraacetic acid (TAE) buffer solution.

To validate the corona formation of hCSF-PNAC, we cross-checked the amyloid corona formation of hCSF-PNAC (Fig. R5a) and PBS-PNAC by HRTEM (Fig. 1c). In both the images, each AuNP was covered with an amyloid corona of uniform thickness (~ 3 nm), which corresponded to the size of a single A β oligomer.

To compare the conformational characteristics of hCSF-PNAC and PBS-PNAC, we used graphene sensors wherein each surface of the sensor was functionalized with the three different 6E10, A11, and OC antibodies. Before the assay using hCSF-PNACs, we checked the affinities between the antibodies and bare AuNPs as a negative control. The relative resistance changes, which represent the affinity between the bare AuNPs and the antibodies, are negligible from all antibodies (Fig. R5b). However, with hCSF-PNACs, the relative resistance values of the 6E10- and A11-immobilized sensors are significantly changed by 2.77% and 3.25%, respectively, implying that the 6E10 and A11 antibodies have strong affinities with the hCSF-PNACs. In contrast, the relative resistance value of the OC-immobilized sensor is unchanged, remaining similar to that of the sensor treated with AuNPs, meaning that the OC antibodies do not capture hCSF-PNACs. These results indicate that the amyloid corona comprises not A β fibrils, but A β oligomers. Altogether, these experiments confirm that the amyloid corona of the hCSF-PNACs comprised A β oligomers, as did those of the PBS-PNACs.

Moreover, we performed gel electrophoresis to analyze the uniformity of the hCSF-PNACs compared to that of the PBS-PNACs (Fig. R5c). The narrow and clear band of the hCSF-PNACs was similar to that of the PBS-PNACs. This result indicated the uniform fabrication of the hCSF-PNACs.

Figure R6. Kinetic analysis of protease XIV using hCSF-PNACs. **a**, UV-vis spectra of hCSF-PNAC solutions depending on protease XIV concentrations of 0, 16, 32, 48, 64, and 112 $\mu\text{g mL}^{-1}$. **b**, Plot fitted by the sigmoidal dose-response curve as a function of protease XIV concentration from 0.1 to 5000 $\mu\text{g mL}^{-1}$.

We performed a proteolytic activity test using hCSF-PNACs as shown in Fig. R6. In detail, we added 0, 16, 32, 48, 64, and 112 $\mu\text{g mL}^{-1}$ of protease XIV to the hCSF-PNAC solutions to investigate its A β -degrading activity. The localized surface plasmon resonance (LSPR) peak of the hCSF-PNACs shifts strongly with increases in protease XIV concentration, which indicates that the amyloid corona of hCSF-PNACs are degraded by the activity of protease XIV (Fig. R6a). From the UV-vis spectra, we analyze the change in relative absorbance ($T_1 - T_0$) as a function of the protease XIV concentration using the dose-response model (Fig. R6b). We extracted the values of the half-maximal effective concentration EC_{50} and the maximal efficacy from the sigmoidal dose-response model as below:

$$\Delta \text{Relative absorbance } (T_1 - T_0) = \frac{[C]^{n_H}}{EC_{50}^{n_H} + [C]^{n_H}}, R^2 = 0.97$$

The values of EC_{50} and maximal efficacy are estimated as 35.19 $\mu\text{g mL}^{-1}$ and 1.01 (a.u.), respectively. These results are consistent with those of the PBS-PNACs.

Figure R7. Kinetic analysis of A β -degrading agent (EPPS) using hCSF-PNACs. **a**, UV-vis spectra of hCSF-PNAC solution depending on EPPS concentrations of 0, 1, 3, 5, 10, 20, and 40 mM. **b**, A_{609}/A_{525} of hCSF-PNAC solution depending on EPPS concentration.

In addition to studying the activity of A β -degrading protease (i.e., protease XIV), we measured the efficacy of EPPS, which can degrade all A β species (i.e., A β oligomers and fibrils) into monomeric A β , against the hCSF-PNACs. The UV-vis spectra of the hCSF-PNAC solution in the presence of EPPS (Fig. R7a) shows that the LSPR peak of the hCSF-PNAC solution shifts with increases in EPPS concentration (1–40 mM).

The A_{609}/A_{525} ratio of the hCSF-PNAC solution is logarithmically increased as a function of the EPPS concentration, providing strong evidence of the ability of EPPS to degrade the amyloid corona of the hCSF-PNAC (Fig. R7b). To investigate the pharmacokinetics of EPPS, we extracted the values of the half-maximal effective concentration EC_{50} and the maximal efficacy by the sigmoidal dose-response model as below:

$$A_{609}/A_{525} = 0.2002 + \frac{0.5758}{1 + 10^{-\text{Log}_{10} [\text{EPPS}] - 1.038}}, R^2 = 0.98$$

The values of EC_{50} and maximal efficacy are estimated as 7.038 mM and 0.776 (a.u.), respectively, which are consistent with those of the PBS-PNAC-based assays (see Fig. 4b).

Altogether, these experiments confirmed that hCSF-PNACs have the same characteristics as PBS-PNACs, and that A β -degrading drug screening can be successfully achieved with hCSF-PNACs in the biological environment of hCSF solution.

We have added this information to the revised manuscript (line 6–13 on page 11 and Supplementary Fig. 21–23).

It would be interesting to extend the PNAC assay to detect not just the disaggregation activity, but also the anti-aggregation activity of candidate compounds. This could be achieved by adding the compounds during the formation of the corona.

Our response: We are thankful for your comment. Following your suggestion, we extended the PNAC assay to detect not just the disaggregation activity, but also the anti-aggregation activity of drug candidates during the formation of the corona on PNAC. We used rutin hydrate, which is known as an anti-aggregation agent, to prevent the transformation of the A β monomer into an oligomer or fibril⁴. In detail, we added 5 mM rutin hydrate solution into an aliquot of the A β sample and incubated for 2 h at room temperature. Then, the solution was mixed with concentrated AuNP solution to the final concentration of 1.4 μ M. We removed the AuNP solution supernatant after centrifugation at 10,000 rpm for 15 min and then added 70 μ L of the final AuNP solution to 160 μ L of the mixed A β solution. The mixed solution was incubated for 24 h in a thermomixer under the incubation conditions of 37 °C and shaking at 120 Hz. The final solution was changed to a suitable buffer solution (DW or PBS) for different experimental conditions.

Figure R8. Anti-aggregation activity test of PNACs using rutin hydrate. **a**, HRTEM images of bare AuNPs, **b**, HRTEM images of PNACs. **c**, HRTEM image of PNACs with 5 mM rutin hydrate in deionized water (DW). **d**, HRTEM image of PNACs with 5 mM rutin hydrate in PBS buffer. **e**, The shift in the plasmonic peak of the PNACs with or without 5 mM rutin hydrate in the PNAC fabrication process. **f**, Agarose gel electrophoresis of the PNAC samples with or without the treatment

of rutin hydrate in the 1× TAE buffer solution.

Without the rutin hydrate anti-aggregation agent, the PNAC was fully covered with an amyloid corona (Fig. 8a, b). In contrast, as shown in Fig. R8c, the formation of the amyloid corona on the AuNPs was restricted in the presence of rutin hydrate. This is because rutin hydrate binds to monomeric A β and thus impedes the formation of an amyloid corona on PNAC⁴. For this reason, the steric stability of PNACs with rutin hydrate was significantly decreased, resulting in aggregation in PBS (Fig. R8d). We confirmed the aggregation of PNACs with rutin hydrate by a salt-resistant test (Fig. R8e). The PNACs without rutin hydrate were stable in PBS because the amyloid corona on the PNACs provided steric stability. In contrast, the PNACs with rutin hydrate were unstable, showing a plasmonic peak shift because rutin hydrate hindered the formation of the amyloid corona of each PNAC. We also performed gel electrophoresis as a suitable assay to ensure the anti-aggregation effect of rutin hydrate (Fig. 8f). The PNACs without rutin hydrate exhibit a distinct band that is attributed to the good dispersion of the PNACs, achieved because the amyloid corona provided good steric stability. On the contrary, the band of the PNACs with rutin hydrate is broadened, implying that PNAC aggregation occurs because of their low steric stability arising from the absence of adequate amyloid oligomers on the PNACs.

These results confirmed that the PNAC assay can be extended not just for the disaggregation activity, but also for the anti-aggregation activity of drug candidates during the formation of the corona on the PNACs.

We added this information to the revised manuscript (line 2–8 on page 4 and Supplementary Fig. 7).

Overall, there are many assays to measure anti-aggregation activity of candidate compounds. The novelty would be to introduce a new assay for oligomer-specific compounds. Not enough evidence is presented that the present study achieves this goal.

Our response: We acknowledge and appreciate your comments. Following your valuable advice, we have performed a series of experiments for validation, as described in the above sections, and thoroughly revised our manuscript and supplementary information to include these new results. The revised manuscript and supplementary information also fully discuss these validation experiments.

References

- 1 Stine, W. B., Jungbauer, L., Yu, C. & LaDu, M. J. in *Alzheimer's Disease and Frontotemporal Dementia*, 13-32 (Springer, 2010).
- 2 De, S. *et al.* Different soluble aggregates of A β 42 can give rise to cellular toxicity through different mechanisms. *Nature Communications* 2019 **10**: 1541 (2019).
- 3 Balducci, C. *et al.* Synthetic amyloid- β oligomers impair long-term memory independently of cellular prion protein. *Proceedings of the National Academy of Sciences* 2010 **107**(5): 2295-2300.
- 4 Wang, S.-w. *et al.* Rutin inhibits β -amyloid aggregation and cytotoxicity, attenuates oxidative stress, and decreases the production of nitric oxide and proinflammatory cytokines. *Neurotoxicology* 2012 **33**(3): 482-490.

Reviewer #3 (Remarks to the Author):

The authors should be commended for having carried out a series of careful control experiments.

Unfortunately, these experiments are still not conclusive to show that the aggregates that form on the surface of the AuNPs during the compound screening assay are oligomers. The main evidence that these aggregates are oligomers is that they bind A11 more than monomers and fibrils. With this evidence, one could call these assemblies “A11-binding aggregates”.

The assemblies on the surface are likely to be adsorbed oligomers. Because of their interactions with the surface, they can't grow into fibrils, remaining stuck in a smaller partially disordered assembly flattened on the surface.

A11 may recognise these flattened oligomers because attaching Abeta fragments onto a surface is indeed the way in which this antibody is generated.

I would be in favour of publication if the authors did not claim that their screening assay identifies compounds that specifically target the type of oligomers formed on pathway to aggregation.

The assay may still be used to find compounds that block surface-induced aggregation, although the nature of the AuNP surface is very different from that of the surfaces found in the brain, which makes the assay less relevant.

Response Letter

Reviewer #3 (Remarks to the Author):

The authors should be commended for having carried out a series of careful control experiments. Unfortunately, these experiments are still not conclusive to show that the aggregates that form on the surface of the AuNPs during the compound screening assay are oligomers. The main evidence that these aggregates are oligomers is that they bind A11 more than monomers and fibrils. With this evidence, one could call these assemblies “A11-binding aggregates”.

Our response: Thank you for your comment. We agree that the amyloid aggregates on AuNPs correspond to “A11-binding aggregates”. We specified that the amyloid aggregates on AuNPs have the characteristics of A11-positive oligomeric aggregates in our revised manuscript (lines 4-8 on page 6). Because the A11-positive aggregates were thought to be partially disordered as you mentioned below, we changed the term of “A β oligomer” to ‘oligomeric aggregate’ throughout the entire manuscript. According to your suggestion, the detailed description regarding the oligomeric aggregates was added in our revised manuscript (lines 4-8 on page 6).

The assemblies on the surface are likely to be adsorbed oligomers. Because of their interactions with the surface, they can't grow into fibrils, remaining stuck in a smaller partially disordered assembly flattened on the surface. A11 may recognise these flattened oligomers because attaching Abeta fragments onto a surface is indeed the way in which this antibody is generated.

I would be in favour of publication if the authors did not claim that their screening assay identifies compounds that specifically target the type of oligomers formed on pathway to aggregation.

Our response: We appreciate your comments. We agree with your view that partially disordered assembly of oligomeric aggregates are stuck on AuNP, and those do not follow the pathway of amyloid fibril formation. We added the above description (your suggestion) regarding the characteristic of oligomeric aggregates, in our revised manuscript (lines 4-9 on page 6). Furthermore, the related previous description regarding the “A β oligomer-to-fibril conversion” was totally deleted, and we rewrote the paragraph in our revised manuscript (lines 10-20 on page 6). Accordingly, we do not claim that our screening assay identifies compounds that specifically target the type of oligomers formed on pathway to aggregation.

The assay may still be used to find compounds that block surface-induced aggregation, although the nature of the AuNP surface is very different from that of the surfaces found in the brain, which makes the assay less relevant.

Our response: We are grateful for your comment. We agree with your comment that the PNAC-based assays may be used to find compounds that block surface-induced aggregation (i.e., anti-aggregation effects of compounds) Following your comment, we added this discussion in the revised manuscript (lines 7-9 on page 4;

line 10–12 on page 12).